# Talin force coupling underlies eukaryotic cell-substrate adhesion

Srishti Rangarajan [1], Lena Espeter [1], Hannes C. A. Drexler [2],
Anna Chrostek-Grashoff[1] & Carsten Grashoff [1] ✉

Integrin-mediated cell adhesion and mechanotransduction are considered key innovations in animal evolution. Here, we show that these processes represent a specialization of an evolutionarily conserved force coupling mechanism that originated in unicellular organisms and is mediated by the actin-binding protein talin. In contrast to heterodimeric integrin receptors, talin is widely distributed in unicellular organisms, including amoebae. By comparing the molecular mechanics of talin-A from amoeboid cells with that of mammalian talin-1, we uncover a conserved role for talin in transmitting pN-scale forces, even in unicellular organisms lacking canonical integrin receptors but expressing the functional homologue SibA. Our data indicate that the critical evolutionary steps towards integrin-mediated cell adhesion in metazoan organisms were the specialization of talin as an adaptor protein allowing the activation of integrin receptors, the regulation of biochemical signaling by paxillin, FAK and YAP, and the control of cell adhesion turnover by KANK recruitment. Collectively, these experiments suggest a central but thus far underappreciated role for talin in the evolution of eukaryotic cell-substrate adhesion and force transmission.

Mechanical signals are important drivers of evolutionary processes[1], and the formation of a mechanically robust attachment of cells to the extracellular matrix (ECM) has been crucial for the evolution of Metazoans[2,3]. The first animals emerged over 700 million years ago and this is thought to have been triggered by the mechanically-induced formation of a first primitive digestive organ[4]. This process of specifying and invaginating a multicellular tissue evolved into the complex developmental process known as gastrulation and depends on the interaction of mesodermal cells with ECM components[4,5]. Many other developmental processes crucial in animals, such as the formation of epithelial tissues, rely on the engagement of cells with ECM structures. Furthermore, many processes of cell locomotion during animal development, such as neural crest cell migration, are governed by the engagement of cells with the extracellular environment[5].

All of these highly specific cell-ECM interactions in Metazoans are mediated by a family of transmembrane receptors called integrins. These adhesion receptors function as constitutive heterodimers consisting of an α- and a β-subunit and are found in all animals. Integrins bind extracellular ligands with high specificity and assemble a complex, multimolecular structure on the cytoplasmic side of the plasma membrane[5]. Depending on the integrin receptor subtype expressed, these adhesion complexes perform a variety of functions by activating biochemical signaling cascades, and by modulating the organization and dynamics of cytoskeletal networks in cells[6,7]. This mechano-chemical signaling is mediated by hundreds of proteins that are directly or indirectly associated with the cytoplasmic tail of the integrin receptors[8–10]. The adapter protein talin plays a particularly central role in these adhesion structures because it is an indispensable integrin activator[11–13], and because it is required to establish a physical connection between integrins and the actin cytoskeleton[14–16]. Therefore, the function of talin has always been closely associated with integrin biology[17,18], and, indeed, talin loss in animal models often closely resembles an integrin null phenotype[19–21]. Intriguingly, talin is found in numerous unicellular organisms that lack canonical

[1]University of Münster, Institute of Integrative Cell Biology and Physiology, Münster, Germany. [2]Max Planck Institute of Molecular Biomedicine, Mass Spectrometry Unit, Münster, Germany. ✉e-mail: grashoff@uni-muenster.de

heterodimeric integrins and also do not produce ECM proteins that are typically bound by these receptors[3]. Phylogenetic analyses have revealed the presence of talin proteins not only in early Obazoa but also in the sister clade Amoebozoa suggesting an evolutionary origin in a common Amorphean ancestor[22], long before the emergence of the first animals[23–26].

The mammalian talin protein consists of two major domains that mediate its role as an integrin activator and force transmitter (Fig. 1a and Supplementary Fig. 1a). The N-terminal head domain facilitates the recruitment of talin to the plasma membrane where it binds and activates integrin receptors but also interacts with other proteins like the signaling molecule focal adhesion kinase (FAK). The C-terminal rod domain directly engages the actin cytoskeleton through two distinct actin-binding sites (ABS) and thereby establishes a molecular linkage that can transmit mechanical forces of at least 7–10 piconewton (pN) during cell adhesion[15]. The integrin-talin-actin linkage is required for the maintenance of mechanically resilient cell-ECM adhesions and their ability to sense extracellular matrix stiffness[14,15,27]. In addition, the rod domain of talin has an adapter function and mediates numerous protein-protein interactions with cytoplasmic molecules such as vinculin, paxillin, and KANK, regulating processes such as adhesion strengthening, biochemical signaling, and cell adhesion turnover[12,16,28]. All these functions as an integrin activator, force transmitter and adapter molecule seem to be conserved across Metazoans. The talin protein found in *Drosophila melanogaster*, for instance, plays a similar

mechanical role as observed in mouse and human cells[20,29]. However, the evolutionary origins of integrin-dependent cell adhesion and mechanotransduction remain unknown, as does the role of the ancient talin proteins found in unicellular organisms lacking canonical integrin receptors.

In this work, we show that the force transmitting role of talin is evolutionarily conserved and already present in Amoeba. Using *Dictyostelium discoideum* as a unicellular model organism that lacks canonical integrin receptors[30,31], we demonstrate that talin mechanically couples the plasma membrane with the intracellular actin network by engaging SibA, a transmembrane protein that shares similarities with the β-subunit of metazoan integrin receptors[30]. The mechanical forces born by this SibA–talin–actin linkage are very similar to the forces observed across talin-1 in focal adhesions of mammalian cells. By contrast, amoeboid talin is poorly evolved as an adapter protein and unable to regulate biochemical signaling through paxillin, FAK and YAP, or to control cell adhesion turnover by KANK recruitment. Together, the data suggest a central, evolutionarily conserved role of talin in eukaryotic cell-substrate adhesion and force transmission.

## Results

### The amoeboid talin protein fails to activate mammalian integrins but transmits pN-scale forces

Talin is widely distributed in the clade of Amoebozoa, including the slime mold *Dictyostelium discoideum*[31–33]. Like many other amoebae, *D.*

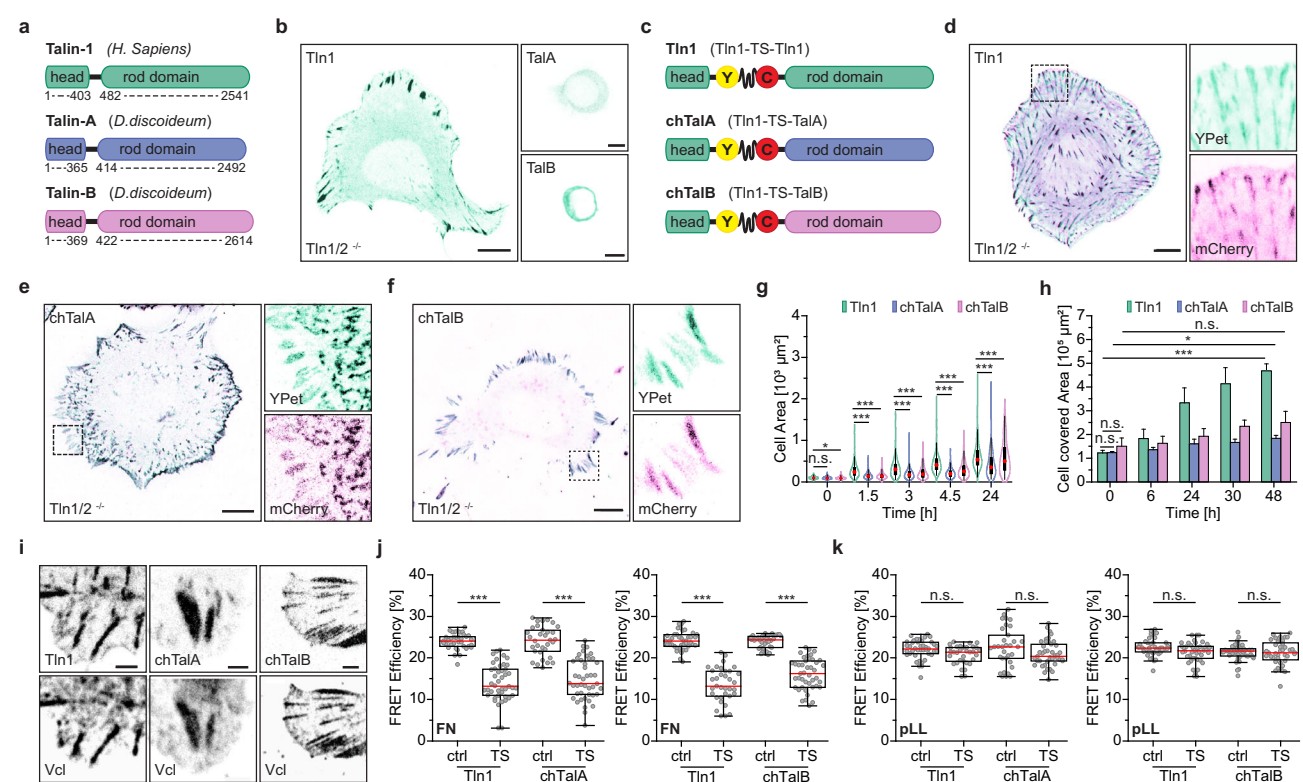

**Fig. 1 | The amoeboid talin rod domain bears mechanical forces. a** Schematic depiction of the domain structure of human Tln1, and TalA and TalB from *D. discoideum*. **b** Representative images of Tln1/2[−/−] fibroblasts expressing either Tln1, TalA or TalB. **c** Schematic illustration of Tln1 or chimeric (chTalA/chTalB) expression constructs. **d–f** Representative live-cell images of Tln1, chTalA and chTalB expressing cells showing FA formation and rescue of cell spreading. **g** Quantification of the cell area at the indicated times of cell spreading. (*n* = 215, 233, 237, 213, 214, 237, 224, 222, 251, 213, 221, 247, 207, 222, 232; *N* = 3). **h** Quantification of cell scratch assays demonstrating defective migration of chTalA/B cells (*n* = 16 per time point, *N* = 4). **i** Vinculin (Vcl) is recruited to FAs in control and chTalA/B cells. **j** Quantification of live-cell FLIM-FRET experiments

reveals mechanical tension across Tln1 and chTalA/B molecules when cells are seeded on FN-coated substrates. (*n* = 37, 44, 33, 44; 42, 35, 42, 43; *N* = 4). **k** FRET efficiency differences between TS samples and no-force controls are lost when cells are cultured on pLL-substrates. (*n* = 41, 33, 34, 42; 43, 43, 43, 40; *N* = 4). Statistical significance indicated in (**g, j, k**) was determined with a Two-sample Kolmogorov–Smirnov test; data in (**h**) were analyzed by a One-way ANOVA; ***$p < 0.001$; *$p < 0.05$; not significant (n.s.) $p > 0.05$. Boxplots show median, 25th and 75th percentile with whiskers reaching to the last data point within 1.5× interquartile range. Scale bars, 10 μm. Source data and exact *p*-values are provided in the Source Data file.

*discoideum* lacks canonical heterodimeric integrin receptors and does not even express individual α- and β-integrin subunits[31]. Therefore, the two *D. discoideum* talin proteins, talin-A (TalA) and talin-B (TalB), with the classical domain structure comprising a defined head and a rod domain (Fig. 1a and Supplementary Fig. 1a), seemed ideal candidates to compare the function of amoeboid and evolutionary more recent talin proteins[34]. We started by investigating codon-optimized TalA or TalB in mammalian fibroblasts genetically depleted of talin-1 and talin-2 (Tln1/2[−/−]). As shown before[15,35,36], lack of talin expression in these cells results in a severe adhesion phenotype caused by the inability to activate integrins and an to form mechanically robust focal adhesions (FAs), effects that are rescued by the re-expression of fluorescently tagged Tln1 or Tln2. However, neither TalA nor TalB expression restored the cell adhesion defect or induced an efficient cell spreading of Tln1/2[−/−] cells (Fig. 1b). As *D. discoideum* does not express canonical integrin receptors, we hypothesized that the head domains of TalA and TalB were unable to bind and activate mammalian integrins. Indeed, we did not observe any β1 integrin receptor activation in Tln1/2[−/−] cells expressing TalA or TalB, while robust signals were present in FAs of talin-deficient cells re-expressing human Tln1 (Supplementary Fig. 1b). We observed isolated dot-like signals in the cytoplasm of Tln1/2[−/−] cells expressing TalA or TalB, but these were also seen in Tln1/2[−/−] fibroblasts (Supplementary Fig. 1c), which are defective in integrin activation[35]. When expressed in the parental Tln1/2[f/f] cells, which express Tln1 and undergo normal FA formation, TalA and TalB remained cytoplasmic (Supplementary Fig. 1d, e). These observations suggest that the head domains of talin proteins from *D. discoideum* are unable to engage mammalian integrin receptors with sufficiently high affinity to allow talin recruitment and integrin receptor activation.

We therefore designed fusion proteins, which comprise the N-terminal head domain of Tln1 and the C-terminal rod domain of TalA or TalB. We reasoned that the expression of such chimeric proteins, referred to as chimeric talin-A (chTalA) and chimeric talin-B (chTalB), should allow for normal integrin receptor activation and cell adhesion, and thus enable us to study the properties of the amoeboid talin rod domains. To visualize these proteins and to facilitate molecular force measurements across talin, we generated constructs in which head and rod domains were connected by a pN-sensitive, FRET-based tension sensor module, following our previously described strategy for talin force measurements[15,36–38] (Fig. 1c). Constructs to enable the quantification of fluorescence lifetimes and the estimation of molecular forces, and to control that observed effects are force-specific, were also produced (Supplementary Fig. 2a). We used our previously established Tln1 tension sensor to record baseline measurements and, as expected, its expression rescued FA formation and cell spreading defects of Tln1/2[−/−] cells[15] (Fig. 1d). The expression of chTalA or chTalB resulted in the induction of cell adhesion and cell spreading (Fig. 1e, f), albeit most cells remained smaller and often failed to maintain a fully spread morphology (Fig. 1g). Chimeric cells which were able to spread displayed prominent peripheral FAs, often close to one another, but formed fewer adhesion complexes in the cell center (Fig. 1e, f and Supplementary Fig. 2b, c). When cells were cultured on soft substrates of 1 kPa, 2 kPa and 8 kPa, this phenotype remained largely unchanged. Control cells spread under all conditions to a larger extent and formed distinct adhesion sites, whereas chimeric cells spread less efficiently and formed narrow rows of FAs (Supplementary Fig. 2d, e). The most striking phenotype, however, was the inability of chimeric cells to undergo mesenchymal migration (Fig. 1h and Supplementary Fig. 2f).

These findings seemed to indicate that the rod domains of TalA and TalB are unable to establish mechanically-resilient force-bearing linkages, which are a prerequisite for maintaining a spread morphology and for efficient cell migration in fibroblasts. However, vinculin seemed normally localized to cell adhesions in both control and chimeric cells (Fig. 1i and Supplementary Fig. 2b). This suggested a mechanical engagement of the chimeric talin molecules, because an efficient recruitment of vinculin to FAs requires a physical extension of the talin rod domain under force[39,40]. To test whether chTalA and chTalB experience such loads, we performed live cell FLIM-FRET experiments using our established protocols[41–43]. Knowing that talin molecules are exposed to comparably high mechanical loads in mammalian cells, we used our HP35-based FRET sensor, which is sensitive to forces of about 6–8 pN[15]. Consistent with our previous studies[15,36], Tln1/2[−/−] cells expressing the Tln1 tension sensor showed markedly reduced FRET efficiencies as compared to a force-insensitive control, and this effect was specific to cells seeded on fibronectin (FN)-coated surfaces (Fig. 1j). When cells were seeded on poly-L-lysin (pLL), which inhibits integrin-specific cell adhesion, differences in FRET efficiencies between control and Tln1 sensor were abolished (Fig. 1k). Intriguingly, molecular forces across chTalA and chTalB were very similar to those observed in Tln1 expressing cells (Fig. 1j) and also lost when cells were cultured on pLL-coated surfaces (Fig. 1k). To confirm that effects were indeed mediated by the talin rod domain, we generated yet another control construct, in which the rod domain was entirely removed. As expected, FRET efficiencies in cells expressing this Tln1 head control were significantly increased to about control levels indicating loss of tension (Supplementary Fig. 2g).

Thus, the rod domains of the amoeboid talin molecules are able to generate a mechanical linkage that resists significant mechanical loads, allowing efficient force coupling during cell-substrate adhesion.

## The amoeboid talin rod does not induce adhesion turnover through KANK

The processes underlying cell spreading and mesenchymal cell migration rely on dynamic cell adhesions, in which FA-resident proteins are exchanged at characteristic rates with molecules in the cytoplasm or the plasma membrane[44,45]. We used fluorescence recovery after photobleaching (FRAP) to test whether the presence of the TalA rod domain alters the turnover of talin molecules in adhesion sites, and we performed these experiments on FN-coated triangular micropatterns to ensure that potential differences were not caused by the distinct morphologies or migratory states observed in control and chimeric cells (Supplementary Fig. 3a). Unexpectedly, the turnover of talin molecules in control and chimeras was very similar, while a Tln1 head control, lacking the entire talin-rod domain, showed significantly faster initial recovery. This suggested that the observed differences in FA morphology and cell migration were not caused by fundamentally altered talin exchange rates (Fig. 2a and Supplementary Fig. 3a, b). We therefore evaluated whether the overall turnover of FAs was altered by performing live cell imaging of cells during mesenchymal migration, and we observed the expected assembly of new and disassembly of old adhesion sites in Tln1-expressing cells (Fig. 2b, Supplementary Fig. 3c and Supplementary Video 1). By contrast, FAs in chimeric cells did not turn over and remained static, in some cases for hours (Fig. 2c, d and Supplementary Videos 2, 3), and new FAs often formed without disassembling neighboring, mature adhesion sites (Fig. 2d).

To identify the underlying cause for this inability of chimeric cells to regulate cell adhesion turnover, we employed mass-spectrometry to determine the interactome of Tln1, chTalA and chTalB, using the Tln1 head domain as a control (Fig. 2e). These experiments confirmed the interaction of Tln1 with known binding partners like vinculin, KANK1[46] and KANK2[47] and also identified a new interactor, the Lipoma-preferred partner (LPP) protein[48], that specifically engages with Tln1 through the rod domain. Intriguingly, LPP was also identified in the interactome, and in FAs (Supplementary Fig. 4a), of both chimeric cell lines, suggesting an evolutionary conservation of the talin–LPP association. By contrast, KANK proteins, which bind the mammalian talin rod at the R7R8 domain[46], were not enriched in chTalA and chTalB interactomes (Fig. 2e), and fluorescently tagged KANK-1 was not recruited to FAs of chimeric cells (Fig. 2f, g). This effect was specific to the rod domain of talin, as replacement of the amino acid sequence in

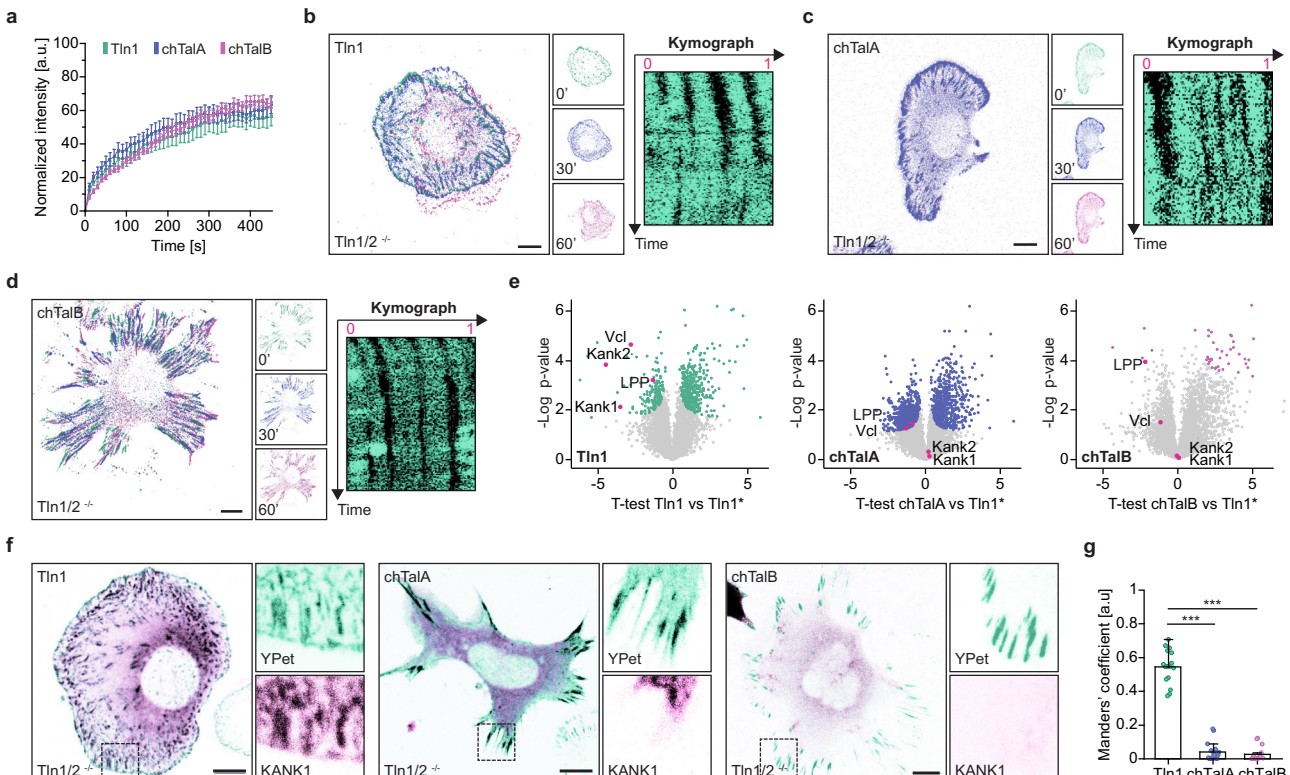

**Fig. 2 | The amoeboid talin rod domain cannot induce adhesion turnover. a** FA turnover rates of Tln1 and chTalA/chTalB are comparable, as revealed by FRAP analysis. (n = 13, 16, 16; N = 4). Error bars indicate s.e.m. **b** Representative image of a cell expressing Tln1, analyzed by time-lapse microscopy. Different colors were assigned to different time points (t = 0, green; t = 30 min, blue; t = 60 min, magenta). The kymograph analysis shows the limited lifetime of FAs in control cells. **c, d** Representative images of cells expressing chTalA or chTalB, analyzed by time-lapse microscopy. (t = 0, green; t = 30 min, blue; t = 60 min, magenta). The kymographs demonstrate that FAs do not disassemble efficiently. **e** Volcano plots of Tln1, chTalA and chTalB interactomes, derived by mass spectrometry. Non-significant proteins are shown in gray whereas statistically significant hits are colored. Note the absence of KANK in chTalA/B samples and the specific enrichment of LPP. **f** Representative images of Tln1/2⁻/⁻ cells co-expressing KANK1-TagBFP with either Tln1, chTalA, chTalB. KANK1 shows efficient FA recruitment in Tln1 cells, but it fails to localize to FAs of chTalA/B cells. **g** Colocalization analysis by quantifying the Mander's Overlap Coefficient confirms defective KANK recruitment in chimeric cells. (n = 16; N = 3). Two-sample Kolmogorov–Smirnov test; *** indicates p < 0.001. Error bars in bar charts indicate the standard deviation (SD) of the mean. Scale bars, 10 μm. Source data and exact p-values are provided in the Source Data file.

the ancient talin rod—corresponding to the R7/R8 domain—with that of mammalian Tln1 was sufficient to recruit KANK back to the vicinity of the FAs (Supplementary Fig. 4b, c). Given the described role of KANKs in regulating FA turnover[49], the inability to localize KANK to adhesion sites seems to explain, at least in part, the impaired FA dynamics in cells expressing the primordial talin rod domain.

### The amoeboid talin rod fails to induce signaling via paxillin, FAK and YAP

Next, we investigated the activation of biochemical signaling which also modulates the turnover of cell-substrate adhesions. While many signaling cascades emanate from FAs and affect a wide range of cellular processes, the phosphorylation of paxillin at tyrosine Y118 and the activation of FAK through phosphorylation of Y397 are closely linked to integrin-mediated cell adhesion, FA turnover, and cell migration[50,51]. Consistent with other studies[14], we have previously shown that FAK phosphorylation at Y397 is markedly reduced in Tln1/2⁻/⁻ cells but gets efficiently rescued upon re-expression of mammalian Tln1[15]. By contrast, expression of either chTalA or chTalB did not elevate FAK pY397 phosphorylation to control levels. Even more striking, the Y118 phosphorylation of paxillin was virtually lost in cells expressing either the chTalA or the chTalB construct (Fig. 3a–c). Activation levels of c-Src and p38 MAPK, two kinases prominently involved in the phosphorylation of paxillin and FAK, were unchanged suggesting a direct role of the mammalian talin rod domain for paxillin/FAK activation.

Examination of cells by immunostaining confirmed our observations and revealed the absence of pY118 paxillin (Fig. 3d) and pY397 FAK (Fig. 3e–g) from cell adhesions of chimeric cells, and even unphosphorylated FAK was poorly recruited (Supplementary Fig. 5a). As talin-1 is a crucial force sensor mediating the translocation of YAP to the nucleus[52], a process that also depends on FAK signaling[53,54], we also investigated whether the subcellular distribution of YAP was altered. Indeed, YAP remained mainly in the cytosol of chimeric cells and its nuclear localization was significantly reduced as compared to Tln1 expressing controls (Fig. 3h, i and Supplementary Fig. 5b).

Collectively, these data suggest that the primordial talin found in *D. discoideum* is capable of mediating a force-bearing mechanical linkage in cell-substrate adhesions. However, it is not yet able to act as an adapter protein that binds and activates canonical integrin receptors, enables adhesion turnover by recruiting KANK proteins, or induces mechanochemical signaling through paxillin, FAK and YAP.

### Talin-A transmits mechanical loads in migrating amoebae

To provide direct evidence that talin does transmit mechanical forces in unicellular organisms, we focused our experiments on TalA, which is the predominant talin isoform in *D. discoideum* at the single-cell, vegetative stage. TalB is most strongly expressed at multicellular stages and was not further analyzed in this study[34]. We used a genetically modified *Dictyostelium* strain lacking TalA (TalA⁻/⁻)[32,33] and a range

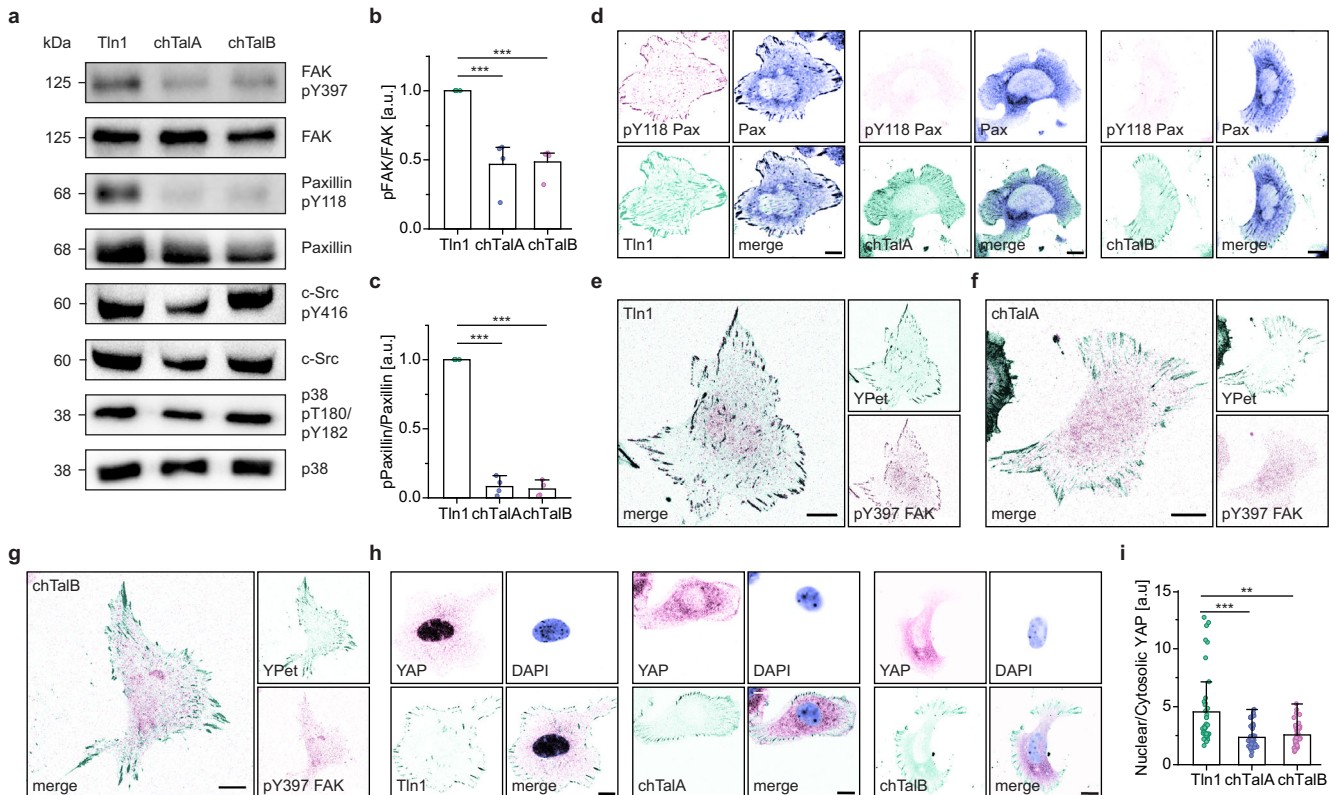

**Fig. 3 | The amoeboid talin-rod does not promote paxillin/FAK/YAP signaling.**
**a** Representative western blots showing reduced pY397-FAK and pY118-paxillin phosphorylation levels in chTalA and chTalB cells. Activation of c-Src and p38 MAPK is unaltered. **b, c** Densitometric quantification of western blot results confirms significant reduction in phosphorylation levels of pY397-FAK and pY118-paxillin in chimeric cells (*N* = 4). **d** Representative immunostaining of paxillin and pY118 paxillin in Tln1, chTalA and chTalB cells. Note that pY118 paxillin is prominent in FAs of Tln1 control cells but entirely absent from FAs of chimeric cells.
**e–g** Representative immunostainings of pY397 FAK in Tln1, chTalA and chTalB cells

showing the absence of pY397 FAK in chimeric cells. **h** Representative images of YAP localization in Tln1, chTalA and chTalB cells co-stained with DAPI to visualize DNA. Note the reduced YAP localization to nuclei in chTalA/B samples.
**i** Quantification of the subcellular distribution of YAP; signals in nuclei were compared to cytosolic values showing reduced nuclear YAP levels in chTalA and chTalB cells. (*n* = 40, 41, 39; *N* = 3). Statistical significance given by One-way ANOVA in (**b, c**) and two-sample Kolmogorov–Smirnov test in (**i**); ***$p < 0.001$; **$p < 0.01$. Error bars indicate the SD of the mean. Scale bars, 10 μm. Source data and exact *p*-values are provided in the Source Data file.

of TalA expression constructs (Fig. 4a) to visualize its subcellular localization and to quantify molecular forces in living amoebae. Expression of a TalA functionality control with a YPet fusion at the C-terminus and expression of the TalA tension sensor both rescued the cell adhesion defect of TalA[−/−] cells (Fig. 4b) and their defective phagocytosis (Fig. 4c, d)[33]; control constructs generated for subsequent FRET experiments showed similar phenotypes (Supplementary Fig. 6a, b). Moreover, the progression of *Dictyostelium* through developmental stages, undergoing cell aggregation, migration, as well as slug and spore head formation, was not adversely affected by the constitutive expression of TalA, which appears to be reduced at late developmental stages under wild type (wt) conditions (Fig. 4e and Supplementary Fig. 6c)[34]. Thus, the addition of single fluorophores or tension sensors between the head and rod domain, or at the C-terminus does not significantly impair the function of the TalA molecule.

*D. discoideum* is light-sensitive and extended live cell FLIM experiments can have an adverse effect on cell shape and motility. Therefore, we utilized an under-agarose, chemotactic migration assay to perform live cell fluorescence microscopy and FLIM-FRET measurements[55]. Under these conditions, amoebae maintained their normal morphology, enabling a detailed analysis of the TalA protein during migration. Consistent with previous reports[56,57], we observed a prominent cortical and posterior localization of TalA. This structure, which often appeared as a dot, was present in cells expressing both control (Fig. 4f) and tension sensor constructs (Fig. 4g) and was

enriched in F-actin (Fig. 4h and Supplementary Fig. 6d). Live cell recordings showed that this talin-rich complex was located in the least mobile part of migrating cells, characterized by a very slow overall turnover, and appeared to attach the cell to the underlying substrate (Supplementary Video 4). Indeed, TIRF microscopy experiments confirmed the close contact of these structures with the glass coverslip (Supplementary Fig. 6e). As previous research has reported the localization of a paxillin homolog, PaxB, to adhesion structures in *Dictyostelium*[58,59], we tested whether such complexes were also formed under our experimental conditions. Using PaxB[−/−] amoeba reconstituted with GFP-PaxB[59], we imaged cells using the under-agarose migration assay, but observed the described, short-lived PaxB accumulations in only a few stationary cells that did not migrate (Supplementary Fig. 6f). Interestingly, however, we also observed dot-like accumulations of PaxB in both stationary and migrating GFP-PaxB expressing cells, which were reminiscent of those observed for TalA (Supplementary Fig. 6f, g).

Given these observations, we hypothesized that TalA acts in the dot-like, actin-rich complexes as an adhesion protein that connects plasma membrane and actin cytoskeleton during cell migration. Since comparatively little is known on the interactome of TalA, we performed mass spectrometry analysis of immunoprecipitates from TalA[−/−] amoebae expressing either the full-length TalA protein or the TalA head domain. These results confirmed the previously described complexation with actin-associated proteins, such as myosin-I heavy

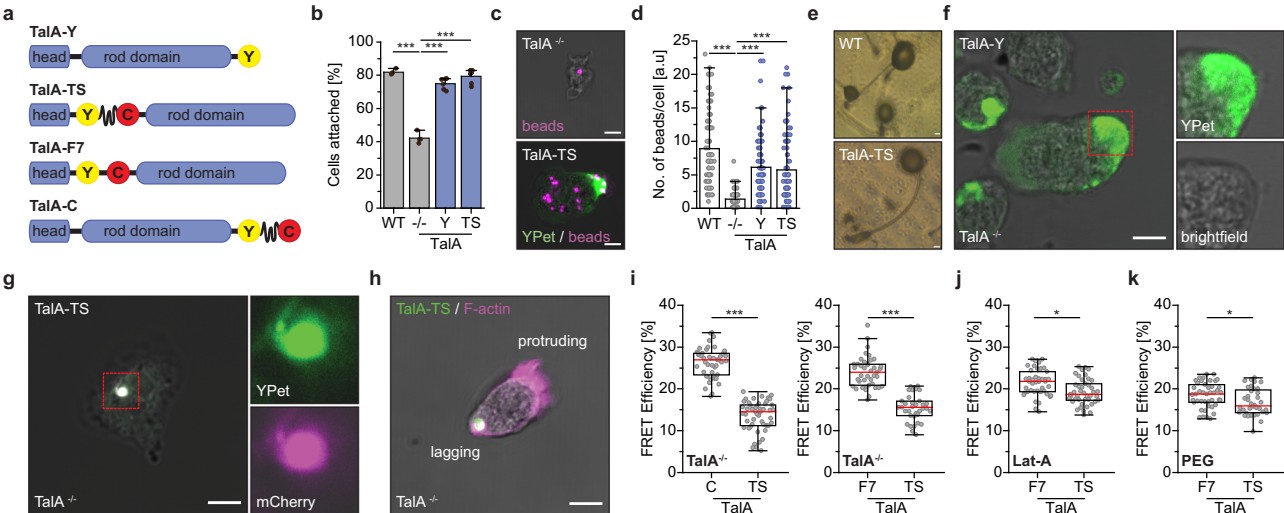

**Fig. 4 | TalA bears mechanical forces in migrating amoeba. a** TalA-Y was used as a functionality control; TalA-TS is the HP35-based TS; TalA-F7 uses a stiff F7 linker and is a no-force control; TalA-C is a second no-force control. **b** TalA-TS and TalA-Y rescue the cell adhesion defect of TalA⁻/⁻ *D. discoideum* cells. (*N* = 6). **c** Representative images of TalA⁻/⁻ and TalA-TS expressing cells with phagocytosed microbeads. **d** The number of beads ingested per cell shows that TalA-TS and TalA-Y expression rescues the phagocytosis defect of TalA⁻/⁻ cells. (*n* = 89, 90, 89, 90; *N* = 3). **e** Images of stalk and spore head formation in wild type (WT) and TalA-TS expressing amoebae showing normal *D. discoideum* development. **f, g** Images of TalA⁻/⁻ cells expressing TalA-Y and TalA-TS, displaying prominent examples of the proximal dot-like structure. **h** Image of a TalA-TS expressing cell revealing co-localization of the talin-rich structure with F-actin. **i** Live-cell FLIM-FRET

measurements of migrating *D. discoideum* cells reveal strongly reduced FRET efficiencies in TS samples, as compared to two independent no-force controls. (*n* = 46, 49; 41, 35; *N* = 4). **j** FRET efficiency differences between no-force controls and TalA-TS cells are reduced after Latrunculin-A (Lat-A) treatment. (*n* = 42, 43; *N* = 3). **k** FRET efficiencies in TalA-TS increases as compared to controls when cells are cultured on pLL-PEG (PEG) surfaces. (*n* = 43, 46; *N* = 3). Statistical significance was tested with a two-sample *t*-test in (**b**), a One-way ANOVA in (**d**) and a Two-sample Kolmogorov–Smirnov test in (**i**–**k**); ***$p < 0.001$; *$p < 0.05$. Error bars indicate SD. Boxplots show median, 25th and 75th percentile with whiskers reaching to the last data point within 1.5× interquartile range. Error bars in bar charts indicate SD of the mean. Scale bars, 50 μm (**e**), 5 μm (**c**, **f**–**h**). Source data and exact *p*-values are provided in the Source Data file.

chain[60] and α-actinin A[61], and revealed an enrichment of additional actin and myosin isoforms such as Actin-3, MyoK and MyoB; interestingly, vinculin was not specifically enriched in these samples. As with mammalian Tln1, however, interactions with the actomyosin network seemed primarily mediated by the TalA rod domain (Supplementary Fig. 7a).

To assess mechanical loading of TalA, we performed live cell FLIM imaging of migrating TalA⁻/⁻ amoebae reconstituted with tension sensor or control FRET constructs. Consistent with the experiments in Tln1/2⁻/⁻ fibroblasts expressing chimeric talin proteins, we detected molecular forces using the 6–8 pN sensitive HP35 sensor, as indicated by a specific FRET efficiency decrease in TalA tension sensor expressing cells compared to two separate controls (Fig. 4i). To confirm this finding, we treated amoebae with latrunculin-A to prevent actin polymerization, which resulted in an increase in FRET efficiency specifically in tension sensor expressing cells, indicating loss of tension (Fig. 4j). Furthermore, when cells were seeded on passivated polyethylene glycol (PEG) surfaces, where cells cannot adhere and TalA accumulates in the cytoplasm, FRET efficiency differences between tension sensor and control samples were largely abolished (Fig. 4k). To test whether TalA would be even exposed to higher forces, as shown for mammalian Tln1[15], we generated TalA tension sensor constructs using the HP35st sensor module (st.TS), which reports forces of 9–11 pN. Amoeba expressing this construct also exhibited reduced FRET efficiencies compared to control cells, suggesting that molecular forces are indeed comparable to those observed across mammalian Tln1 (Supplementary Fig. 7b).

Taken together, these data demonstrate that TalA is subject to mechanical loads in migrating *D. discoideum*, despite the absence of canonical heterodimeric integrin receptors. Remarkably, the magnitudes of molecular forces experienced by talin in mammalian cells and amoebae are very similar[15,36].

## Talin-A engages alternative transmembrane receptors for force coupling

In amoebae, TalA is thought to connect to the plasma membrane by associating with SibA, a transmembrane protein comprising two NPXY motifs in the cytoplasmic tail of the molecule[30]. Owing to this similarity and the presence of a von Willebrand factor type A domain in the extracellular region, which is also found in the ectodomain of integrins, SibA is considered a functional equivalent of β-integrins. To test whether SibA is required for TalA-mediated force transmission, we expressed the TalA tension sensor and control constructs in SibA-deficient cells (SibA⁻/⁻) but also in the parental DH1-10 cells to account for potential over-expression artefacts. In wt cells, FLIM-FRET experiments yielded similar results as observed in reconstituted TalA⁻/⁻ cells: high FRET values in control cells and reduced FRET efficiencies in TalA-TS expressing cells, showing force transmission across TalA. When the same constructs were expressed in SibA⁻/⁻ amoebae, TalA still assembled in the actin-rich, dot-like adhesion structure but it was not under tension (Fig. 5a). To confirm these observations, we performed TalA force measurements in SadA-deficient cells (SadA⁻/⁻). SadA is a transmembrane protein that is essential for cell adhesion in *Dictyostelium* at least in part by controlling SibA expression on the transcript and the protein level[62,63]. Again, we observed forces acting on TalA in the parental AX3 strain but a loss of TalA forces in SadA⁻/⁻ cells (Fig. 5b). These data suggest that TalA can engage alternative, integrin-like cell surface receptors to mediate force transmission between the plasma membrane and the interior of the cell, presumably the actin cytoskeleton[56,64].

To provide direct evidence that force transmission also in amoebae relies on the talin rod domain and its interaction with actin, we generated TalA constructs, in which the C-terminal domain ended after amino acid 940 (TalA*) to remove all

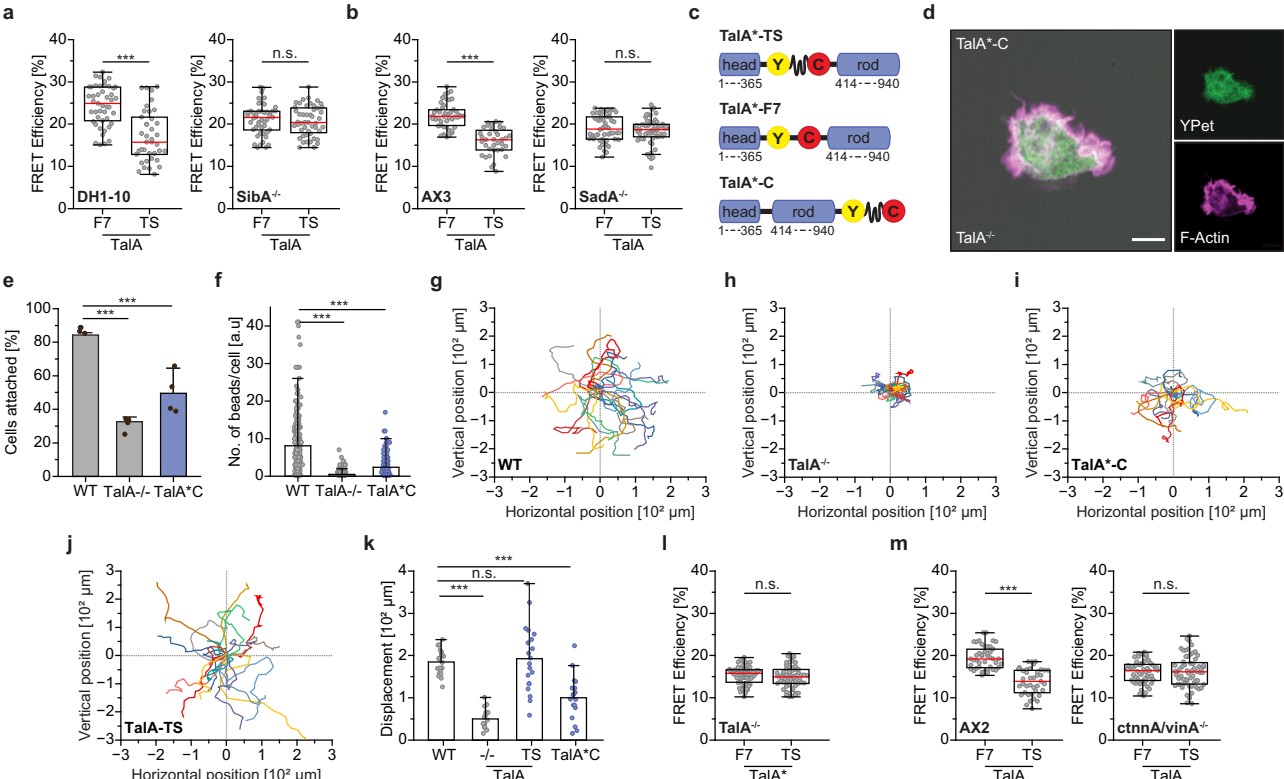

**Fig. 5 | TalA mediates force coupling between SibA and the actin network.**
**a** FLIM-FRET measurements of migrating *D. discoideum* cells; TalA-TS and the TalA-F7 control were expressed in SibA$^{-/-}$ cells and the parental line, DH1-10. (*n* = 45, 39; 47, 48; *N* = 4). **b** FLIM-FRET measurements in SadA$^{-/-}$ cells and the parental line, AX3. (*n* = 48, 38, 52, 57; *N* = 3). **c** Schematic depiction of truncation mutant (TalA*) constructs. **d** Representative image of a TalA$^{-/-}$ cell expressing a TalA* construct and stained for F-Actin. **e** TalA* expression does not rescue the cell adhesion defect of TalA$^{-/-}$ cells. (*N* = 4). **f** TalA*-C expression fails to rescue phagocytosis defects. (*n* = 242, 316, 128; *N* = 3). **g–j** Trajectory plots of migrating wild type (WT), TalA$^{-/-}$, TalA*-C and TalA-TS cells; the truncation construct does not rescue cell migration defects of TalA$^{-/-}$ amoebae. (*n* = 20; *N* = 3). **k** Quantification of cell displacement in

WT, TalA$^{-/-}$, TalA-TS and TalA*C cells. (*n* = 20, 16, 20, 18; *N* = 3). **l** FLIM-FRET measurements of TalA$^{-/-}$ cells expressing TalA*-TS or the TalA*-F7 control indicating the loss of mechanical tension across talin. (*n* = 62, 62; *N* = 4). **m** FLIM-FRET measurements in ctnnA/vinA deficient cells and the parental line, AX2. Talin forces are lost in the absence of vinculin expression. (*n* = 45, 39; 58, 64; *N* = 4). Statistical significance was determined with a two-sample t-test in (**e**), a One-way ANOVA in (**f**) and (**k**) and a Two-sample Kolmogorov–Smirnov test in (**a**, **b**, **l**, **m**); ***$p < 0.001$; *$p < 0.05$. Boxplots show median, 25th and 75th percentile with whiskers reaching to the last data point within 1.5× interquartile range. Error bars in bar charts indicate SD of the mean. Scale bar, 5 μm. Source data and exact *p*-values are provided in the Source Data file.

potential ABSs (Fig. 5c). As a result of this truncation, the characteristic talin-rich structure was lost in TalA$^{-/-}$ cells expressing these truncated proteins (Fig. 5d), the cells still showed prominent cell adhesion (Fig. 5e) and phagocytosis defects (Fig. 5f), and they did not migrate as efficiently as the wt and TS controls (Fig. 5g–k). Most importantly, the truncation of the rod domain resulted in a loss of molecular tension across the TalA molecule (Fig. 5l). As force coupling through Tln1 is modulated by vinculin in mammalian cells[15,36], we also investigated the potential role of a ctnnA/vinA, an amoebic protein that has been proposed to be a precursor molecule of both vinculin and α-catenin and appears to function in both cell-matrix and cell-cell adhesion[65,66]. While we observed the expected mechanical loading in the parental cells, FRET differences between controls and TalA-TS cells were abolished in the absence of ctnnA/vinA (Fig. 5m). As noted above, we did not observe ctnnA/vinA in our TalA interactome, thus future experiments should test whether the regulation of TalA tension through ctnnA/vinA is indirect or mediated through transient, potentially force-stabilized interactions that are difficult to detect in the here performed co-immunoprecipitation studies.

Overall, these data demonstrate that the function of talin as a mechanical linker, connecting cell surface receptors to the actin cytoskeleton, is already established in amoebae and efficiently operates in the absence of heterodimeric integrin receptors.

## The SibA-TalA linkage is sufficient to promote amoeboid migration in mammalian cells

Our data demonstrate that the talin molecule, which is found in unicellular organisms, is already capable of coupling membrane receptors with intracellular actin networks. However, additional features, including high-affinity integrin binding, KANK recruitment, and biochemical signaling via paxillin, FAK and YAP, seem to have been acquired throughout evolution to orchestrate integrin-mediated cell adhesion in metazoan cells[67]. Considering these developments, we wondered whether the original, mechanical function of talin is still sufficient to modulate processes in mammalian cells. We therefore reconstituted Tln1/2$^{-/-}$ fibroblasts with SibA and TalA together, or only with SibA as a control. Similar to the preceding experiments, cells maintained their round morphology and were not rescued by the expressed proteins under normal 2D culture conditions, neither on stiff nor on soft substrates (Supplementary Fig. 8a–d). However, confining cells to a height of 5 μm induced a notable phenotype. Previous studies have shown that this type of cell confinement induces cell blebbing and amoeboid cell migration, if integrin-dependent cell adhesion is also supressed[68]. As expected, the Tln1/2$^{-/-}$ control cells, which lack active integrin receptors, started to bleb and individual cells transitioned into a migratory mode upon confinement (Fig. 6a and Supplementary Video 5). Intriguingly, Tln1/2$^{-/-}$ cells expressing only SibA showed no induction of cell migration under cell confinement

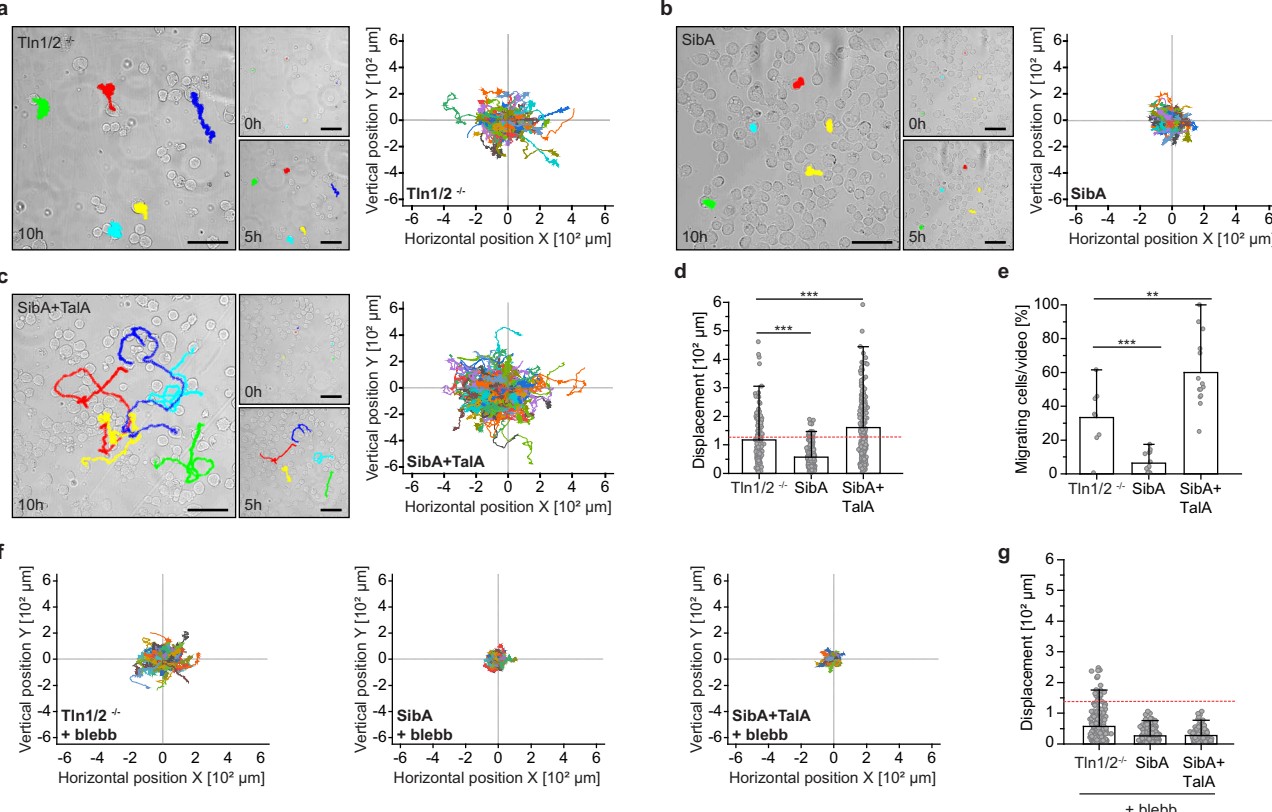

**Fig. 6 | The SibA-TalA linkage promotes amoeboid migration in fibroblasts.**
**a** Representative images from time-lapse videos showing Tln1/2⁻/⁻ cells at the beginning of the experiment, after 5 h and 10 h of confinement. Cumulative tracks of five representative cells are highlighted in colors. The trajectory plots of all tracked cells are shown on the right. (*n* = 131; *N* = 3). **b** Representative images from time-lapse videos of Tln1/2⁻/⁻ cells expressing SibA. Shown are the images at the beginning of the experiment, after 5 h and 10 h of confinement. Cumulative tracks of five representative cells are highlighted in colors. The trajectory plots reveal that the expression of SibA alone inhibits amoeboid migration in fibroblasts. (*n* = 329; *N* = 3). **c** Representative images from time-lapse videos showing Tln1/2⁻/⁻ expressing SibA and TalA at the beginning of the experiment, after 5 h and 10 h of confinement. Cumulative tracks of five representative cells are highlighted in colors. The

trajectory plots reveal that reinstating the SibA-TalA linkage increases amoeboid migration under confined conditions. (*n* = 228; *N* = 3). **d** Quantification of the overall displacement of cells under confinement. A cumulative displacement of less than 120 µm over 10 h was considered non-migratory. (*n* = 131, 352, 228, *N* = 3). **e** Quantifying the number of migrating cells reveals significantly increased amoeboid migration in cells expressing SibA and TalA. (*n* = 8, 14, 14; *N* = 3). **f** Treating cells with para-amino blebbistatin (blebb) strongly reduces amoeboid cell migration. **g** Quantification of the overall displacement of cells under confinement in the presence of para-amino blebbistatin (blebb) (*n* = 225, 330, 157; *N* = 3). Statistical significances were determined with a One-way ANOVA: ***$p < 0.001$; **$p < 0.01$. Error bars in bar charts indicate SD of the mean. Scale bar, 50 µm. Source data and exact *p*-values are provided in the Source Data file.

and, in fact, remained stationary (Fig. 6b and Supplementary Video 6). By contrast, cells co-expressing SibA and TalA frequently transitioned to an amoeboid, migratory phenotype (Fig. 6c and Supplementary Video 7) resulting in a significantly higher overall displacement (Fig. 6d) and an increased number of migrating cells (Fig. 6e). At this resolution, we did not observe a notable change in the subcellular localization of TalA when cells were migrating (Supplementary Fig. 8e). However, treating cells with para-amino blebbistatin entirely abrogated migration suggesting that it is driven by a myosin-II dependent process, as described before[68] (Fig. 6f, g).

These data show that the talin protein from *Dictyostelium* can promote mechanical coupling in mammalian cells when a suitable membrane adapter such as SibA is provided, and they indicate that this talin-dependent force coupling can promote amoeboid migration of mammalian cells.

## Discussion
Little is known about the evolution of integrin-dependent cell adhesion and mechanotransduction. Although homologs of a growing number of metazoan adhesion proteins are being identified in unicellular organisms[23–26,31], we have limited insight into their function and do not yet understand how they have evolved to synergize and

orchestrate the complex mechanical and biochemical signaling observed in animals. Here, we show that the intracellular machinery for transmitting mechanical forces during cell-substrate adhesion is highly conserved and already present in slime molds. Central to this early form of cell-substrate dependent force transmission is the talin protein, which is best known for its role as an integrin activator in animals[11,13,17] but which is also widely distributed in unicellular organisms, including those lacking canonical α/β-integrin receptors[24,31]. It is an open question whether talin evolved before or at the same time as α- and β- integrins, which are found in some, but not all amoebozoan genomes[31]. However, the fact that few Amoebozoans express both α- and β-integrin subunits[31] suggests that these early transmembrane proteins may not yet function as the heterodimeric integrin receptors found in mammalian organisms[67]. Regardless of the precise phylogenetic origins, our experiments show that force transmission across the ancient talin protein does not require the presence of heterodimerc integrin receptors, known from Metazoans. Instead, talin from *Dictyostelium* anchors to the plasma membrane through functionally equivalent transmembrane proteins, such as SibA, to provide a mechanical link between the plasma membrane and the actin cytoskeleton. Remarkably, the molecular forces relayed across the talin molecule in amoeba are very similar to those observed in flies[29] and

mammalian cells[15,36], and are also dependent on the interaction with the actin cytoskeleton through the C-terminal rod domain of talin[15,36,69]. It therefore appears that force propagation through talin, mediated by an N-terminal engagement to a transmembrane receptor and a C-terminal association with the actomyosin network, is an evolutionarily conserved mechanism that already enabled unicellular organisms to adhere to the substrate and migrate, when canonical integrins—and extracellular matrix components they bind to[3]—were either poorly evolved or entirely absent. The extent to which extracellular forces across those alternative receptors are comparable to the catch-bond mechanics seen across integrin–ECM linkages is one of the exciting questions that should to be examined in the future.

Other talin-specific functions, such as those important for the turnover of FAs, seem to have evolved later. Consistent with other studies[56], we observed a posterior localization of talin in migrating *Dictyostelium* cells which often accumulated in a dot-like adhesion structure on the basal surface of the cell—reminiscent of recently described ventral adhesion dots[70]—where it colocalized with actin. Interestingly, even in migrating *Dictyostelium* cells, these adhesions exhibited very slow, indeed almost absent, turnover: once formed, they remained at the rear, least motile part of the cell, seemingly anchoring it to the underlying surface. This behavior is worth noting because FAs undergo a constant turnover in mammalian cells, particularly during cell migration. Adhesion turnover is known to be complex, but at least partly mediated by KANK proteins, which promote microtubule targeting of FAs leading to their sliding and disassembly[49]. Our data indicate that the amoeboid talin is unable to interact with KANK proteins and indeed, to our knowledge, no KANK genes have been identified in *Dictyostelium*.

Other mechanisms known to modulate adhesion turnover in Metazoans involve biochemical signaling through paxillin and FAK[71], two molecules that are constitutive members of the mammalian integrin-based adhesome[10]. Integrin-dependent cell adhesion modulates the activity of these molecules by phosphorylation of distinct tyrosine residues, namely FAK-pY397 and paxillin-pY118, leading to the initiation of a wide range of signaling pathways that can affect the transcriptional profile of cells by modulating the nuclear localization of YAP. Our experiments demonstrate that these processes require full length Tln1 and cannot occur when the rod domain is replaced with that of the amoeboid talin. These results seem consistent with the fact that FAK is absent from Amoebozoa and emerged later during the evolution of eukaryotes[24,59,72]. *Dictyostelium* expresses paxillin-like proteins, and especially PaxB shares high homology with the mammalian protein. However, the characteristic phosphorylation target residues, Y118 and pY31, are not conserved in the amoebozoan paxillin. Moreover, unlike TalA, the amoebic PaxB is predominantly expressed at multicellular developmental stages[58]. We did notice a dot-like accumulation of GFP-PaxB in migrating amoeba that was reminiscent of the punctate talin structures, but it still needs to be tested whether both proteins colocalize in these complexes. It is also worth noting that paxillin, like vinculin, was not specifically enriched in our *Dictyostelium* TalA interactome, whereas previously identified interaction partners such as MyoI were present. All this indicates that central mechanisms known to regulate integrin-dependent signals in animals are not yet fully established in unicellular organisms, even though the individual proteins may already be present. This makes it all the more remarkable that the mechanical role of talin is so similar in *Dictyostelium* and mammals. While available phylogenetic evidence suggests a direct evolutionary path from TalA to Tln1[24], it remains to be clarified how SibA and mammalian integrin receptors are related[30]. Therefore, it should be fascinating to conduct experiments on additional single-celled eukaryotic species to determine the underlying evolutionary mechanism, and investigate the potential for processes of convergent evolution.

Given the expression of talin in numerous unicellular Amorphea[24,26], it is tempting to speculate that talin force coupling is not only a conserved but also a widely used mechanism in single-celled eukaryotes to enable cell attachment, cellular locomotion, and phagocytic processes. Given the pathological relevance of numerous organisms that may rely on talin force coupling, further studies seem warranted. *Entamoeba histolytica*, for example, infects millions of people worldwide and kills tens of thousands each year[73]. Similar to *Dictyostelium*, this parasite expresses a talin-like molecule but lacks the classical integrin heterodimers and mediates cell adhesion through an alternative receptor. This transmembrane receptor, a lectin, is crucial for the virulence of the parasite and its cytoplasmic tail shares homology with β2 integrins[74] suggesting that it may indeed bind talin. Similarly, some *Acanthamoeba*, which can cause debilitating eye infections, also express a talin protein, lack integrins receptors and use a lectin, the mannose-binding protein (MBP), for adhesion to the surface of the cornea[75]. Again, the MBP receptor is a crucial virulence factor and is characterized by an integrin-like cytoplasmic domain containing a potential talin-binding NPXY motif[75]. In light of our findings, it should be tested whether these receptors, similar to SibA in *Dictyostelium*, use the respective talin protein for integrin-independent force coupling and the formation of mechanically resilient cell adhesions.

Interestingly, this ancestral function of talin may not only be relevant in unicellular organisms. Our reverse evolution experiments show that the restoration of integrin-independent force coupling in mammalian fibroblasts is indeed sufficient to stimulate amoeboid migration when the cells are confined. The intriguing question is whether mammalian talin proteins can still use alternative transmembrane receptors to mediate force coupling. Certainly, such a mechanism would not compensate for the loss of integrin-mediated processes, but rather provide an alternative mode of mechanically connecting the plasma membrane to the force-generating actin network when activated integrins are not available - either because they are expressed at low levels or because they are unable to bind extracellular ligands. Addressing this question may indeed be relevant to integrin-independent modes of immune cell migration and cancer cell dissemination[71–73].

Regardless of how these intriguing questions are answered, it appears that we have to reconsider the role of talin for the evolution of eukaryotic cell adhesion and mechanotransduction. Our data suggest that talin was critical as a force coupling molecule for eukaryotic cell-substrate adhesion long before the first animals evolved. We propose that integrin-mediated cell adhesion, as it operates in animals, is a specialization of this primordial talin-based mechanism.

## Methods
### Antibodies
The following primary antibodies were used for immunofluorescence staining (IF) and Western blotting (WB): anti-talin (TA205, Bio-Rad, MCA725G; WB: 1:2000), anti-vinculin (Sigma-Aldrich, V9131; IF: 1:400), anti-p-S19-myosin light chain 2 (Cell Signaling, 3671; IF: 1:50), anti-LPP (Abcam, ab126608; IF: 1:200), anti-tubulin (DM1A) (Sigma-Aldrich, T6199; WB: 1:5000), anti-FAK (Millipore, 06-543; IF: 1:200, WB: 1:2000), anti-pY397-FAK (Invitrogen, 44-624; IF: 1:200, WB: 1:1000 in 5% BSA), anti-Paxillin (BD Transduction Laboratories, 610051; IF: 1:200; WB: 1:1000), anti-pY118-Paxillin (Millipore, 07-1440; IF:1:200; WB:1:1000), anti-GFP (Abcam, ab290; WB: 1:2000),anti-YAP (63.7) (Santa Cruz, sc-101199; IF:1:400), anti-Integrin β1 (9EG7) (BD Pharmingen, 550531; IF: 1:200). The following secondary antibodies were used: anti-mouse IgG Alexa Fluor 647 (Thermo Fisher Scientific, A21235; 1:200), anti-rabbit IgG Alexa Fluor 647 (Thermo Fisher Scientific, A21244; 1:200), anti-mouse IgG Alexa Fluor 405 (Thermo Fischer Scientific, A48255; 1:500), anti-rat IgG Alexa Fluor 647 (Thermo Fischer Scientific, A21247; 1:500) anti-mouse IgG HRP (Bio-Rad, 170-6516; 1:10000), anti-rabbit IgG HRP

(Bio-Rad, 170-6515;1:10000). Alexa Fluor 647 Phalloidin (Thermo Fisher Scientific, A22287; 1:200) was used to visualize f-actin. DAPI (Sigma, D8417; 1:1000) was use to stain the nucleus.

## Generation of talin expression constructs

The talin-1 expression construct was based on human talin-1 cDNA (accession number: BC042923) and the YPet-HP35-mCherry tension sensor (TS) module (Addgene, 101250), described previously[15]. For FRET controls, a donor-only lifetime construct was generated by inserted a point mutation into mCherry (Y72L) of the TS module; the first no-force control used a non-stretchable F7 linker[76] instead of the mechanosensitive HP35 sequence. The second no-force control was generated by fusing the TS module to the C-terminal end of the respective talin construct. For the expression of TalA (UI14576), TalB (Q54K81) and SibA (Q54KF7) in mouse fibroblasts, cDNA sequences were codon-optimized using Genewiz services (Azenta Life Sciences). All sequences were assembled in the mammalian expression vector pLPCX (Clontech, 631511) using NEBuilder® HiFi DNA Assembly (New England Biolabs), and the correct sequence was confirmed with DNA Sequencing (Microsynth Seqlab).

For the generation of TalA tension sensor constructs, NEBuilder® HiFi DNA Assembly (New England Biolabs) was used to insert the YPet-HP35-mCherry into TalA (a gift from M. Tsujioka, Tokyo Medical and Dental University) after aa 405; sequences were assembled in pDEX-RH (a gift from J. Faix, MHH Hanover). The first no-force control used a non-stretchable F7 linker[76] instead of the mechanosensitive HP35 sequence. The second no-force control was generated by fusing the TS module to the C-terminal end of the respective talin construct. As a functionality control, TalA was C-terminally fused with YPet. TalA truncation constructs were terminated at aa 940. Correct cDNA sequences were confirmed by DNA sequencing (Microsynth Seqlab).

## Cell culture and expression of cDNA constructs

Mammalian fibroblasts were cultured in high glucose DMEM-GlutaMAX medium (Thermo Fisher Scientific) supplemented with 10% fetal bovine serum (FBS; Thermo Fisher Scientific) and 1% penicillin/streptomycin (P/S; Thermo Fisher Scientific). For stable expression, cDNA constructs were transduced by the Phoenix cell transfection protocol[43,77] and selected with Puromycin (1 μg/mL; Sigma). Transduced cells were sorted by fluorescence-activated cell sorting using a FACS-Aria III cell sorter to ensure comparable expression levels.

The following *D. discoideum* strains were used as indicated: AX2 (gift from J. Faix, MHH Hanover), TalA[−/−] [33], SibA[−/−] [60], DH1-10 (gift from P. Cosson, University of Geneva), VinA[−/−] (gift from G. Gerisch, MPI Martinsried), SadA[−/−] [63], AX3 and PaxB[−/−] or AX2 stably expressing PaxB-GFP[58]. Cells were grown axenically in HL5-C medium including glucose (Formedium) supplemented with 0.05 mg/mL Ampicillin and 0.04 mg/mL Streptomycin at 22 °C in an incubator with a light source. Phosphate buffer (PB; 2 mM Na$_2$HPO$_4$. 2H$_2$O, 15 mM KH$_2$PO$_4$, pH 6.0) was used to wash the cells. TalA-TS constructs were transfected in the indicated strains by electroporation and transfected cells were selected on a lawn of *Klebsiella aerogenes* cultured on 1.5% Phosphate agar plates with 50 μg/mL G418 antibiotic at 22 °C. Individual colonies were expanded in HL5-C glucose media containing 10 μg/mL G418.

## Fixation and Immunostaining

For immunostainings, 2 ×10$^4$ fibroblasts were seeded onto 35 mm ibitreat μ-dishes (ibidi), allowed to spread overnight, fixed with 4% paraformaldehyde (PFA; Roth), and washed 3 times with phosphate-buffered saline (PBS) (Sigma, D8537-6X500ML). Cells were then incubated in the blocking buffer containing 2% Bovine Serum Albumin (BSA; Serva) and 0.1% TritonX-100 (Roth) for 1 h at RT. Primary and secondary antibodies were diluted in the blocking buffer and incubated overnight at 4 °C and 1.5 h at RT. *D. discoideum* cells were seeded

in an 8-well chamber slide (ibidi) 2–3 h before fixation. The cells were washed with Phosphate buffer (PB) and fixed for 30 min at RT in the dark using 15 % Picric acid (Sigma) PFA in 10 mM PIPES (Sigma) at pH 6.0[78]. Cells were washed twice with PIPES buffer and twice with PBS/Glycine (0.75%; Sigma). For the visualization of actin, cells were permeabilized in 70% Ethanol (Sigma, 32205) for 10 min followed by two washing steps with PBS/Glycine. Next, samples were blocked in PBG buffer (1× PBS, 0.5% BSA, 0.045% Fish Gelatin (Sigma, G7041) for 30 min, phalloidin was diluted in the blocking buffer, and then incubated on the cells overnight at 4 °C.

## Microscopy of mammalian and *Dictyostelium* cells

For live cell imaging of mammalian cells, fibroblasts were seeded onto FN-coated (10 μg/mL) 35 mm glass bottom μ-dishes (ibidi, 81158-400) and allowed to adhere overnight. Before imaging, media was exchanged to phenol red free imaging medium (Thermo Fisher, 21063045) supplemented with FBS and P/S (imaging medium). *D. discoideum* cells were seeded onto 35 mm ibi-treat μ-dishes and allowed to adhere for 2 h. Cells were washed with PB and the medium was exchanged to Low Fluorescence (Lo-flo) medium (Formedium) supplemented with P/S prior to imaging. For visualization of live cells under agarose, a chemotactic gradient was set up using 100 μM Folic acid (Supplementary Fig. 9a). Image acquisition was carried out on a LSM880 confocal laser scanning microscope (Zeiss) equipped with a 63× glycerin objective (LCI Plan-Neofluar 63×/1.3 Imm Korr DIC M27) using ZEN Software, black edition (Zeiss). Live cell imaging of fibroblasts was conducted at 37 °C and *D. discoideum* was imaged at RT. TIRF Imaging for *D. discoideum* cells was performed on fixed samples using a Elyra 7 TIRF microscope (Zeiss) equipped with 63× oil objective (alpha Plan-Apochromat 63×/1.46 Oil Korr M27 Var2). The mean fluorescence intensity of the desired region (FA, nucleus or cytoplasm) was determined using Fiji. Colocalization was quantified using Manders' overlap coefficients (M1, M2) in Fiji (Coloc2 plugin).

## Wound healing assay, cell area analysis, and FA quantification

4 ×10$^4$ cells were seeded into 2-well silicon culture inserts (ibidi) on 35 mm glass bottom dishes coated with FN. Cells were allowed to adhere overnight and media was supplemented with 5 μM Cytosine Arabinoside (AraC) 4 h before the start of imaging to inhibit proliferation. For imaging, medium was exchanged to imaging medium and images were acquired at the indicated time points with an LSM880 confocal microscope (Zeiss) using a 20× objective (Plan-Apochromat 20×/0.8 M27). For quantifying the cell area, fibroblasts fixed at the indicated time points were incubated for 30 min at RT with CellMask Deep Red plasma membrane stain (Thermo Fisher Scientific, C10046). Images were acquired with a Zeiss LSM880 microscope using a 20× objective. For determining FA number and area, fixed samples were imaged at 63× magnification. Using the published protocols[79], images were analyzed in Fiji with the following parameters: subtract background (sliding paraboloid, radius=20 pixels); CLAHE (block size = 19, histogram bins = 256, maximum slope = 6) for enhancing contrast; EXP to minimize the background; Brightness & Contrast (Auto); median filter (radius = 3 pixels); Threshold (Shanbhag method); Analyze particles (size = 0.3 μm$^2$−infinity, circularity = 0.00−0.99, exclude on edges). Generated ROI were verified with the original fluorescent image.

## Fluorescence recovery after photobleaching (FRAP)

FRAP experiments were performed on FN-coated Y-shaped micro patterned substrates (CYTOO) following previous protocols[15,36,76]. In brief, 4 ×10$^4$ cells/mL were seeded and allowed to spread overnight. Using a Zeiss LSM880 microscope and a 63× glycerin objective, three pre-bleach images were recorded to establish a baseline fluorescence value, followed by bleaching of selected FAs at 514 nm. Fluorescence recovery was then recorded for 300 s at 10 s intervals. The data was

imported into ImageJ for extracting the intensities of ROI, background and pre-bleach images and corrected for background intensities.

Using Jay Plugins (https://research.stowers.org/imagejplugins/zipped_plugins.html), normalized FRAP curves were plotted in OriginPro and fitted according to Eq. (1), where ($t$) represents time, ($b$) the rate constant, and ($a$) the mobile fraction.

$$f(t) = a\left(1 - e^{-bt}\right) \tag{1}$$

### Cell adhesion and phagocytosis assay, analysis of developmental stages

For assessing the cell-substrate adhesion, $1 \times 10^6$ *D. discoideum* cells were resuspended in 2 mL of KK2 buffer (16.5 mM $KH_2PO_4$ and 3.8 mM $K_2HPO_4$, pH 6.2; Roth) and seeded in a 60 mm petri dish. Cells were allowed to adhere for 30 min at RT and then agitated at 75 rpm for 60 min on a gyration shaker. The number of unattached cells in the supernatant was quantified and used to calculate the percentage of attached cells (Supplementary Fig. 9b). For phagocytosis assays, $1 \times 10^6$ cells/mL were seeded into a 4-well chamber slide (ibidi, 80426) and agitated for 2 h at 150 rpm on a gyration shaker. Next, cells were incubated with 10 μL of 1.0 μm polystyrene beads, diluted 1:5 in Lo-flo medium for 30 min at 120 rpm. The chamber was washed twice with Lo-flo medium and fixed using the PFA-Picric acid protocol. The number of engulfed beads per cell was determined by confocal imaging using a 63× glycerin objective (Supplementary Fig. 9c). For investigating developmental stages, $1 \times 10^7$ cells in vegetative state were washed and resuspended in 500 μL PB, plated onto 1.5% phosphate agar plates and incubated at 22 °C for 2 days. Cell aggregation, spore head and stalk formation were monitored with a Leica DM IL LED microscope using a 10× objective (HI PLAN CY 10×/0.25 PH1). Images were captured with of a Leica MC170 HD camera using the LAS EZ software.

### Random migration and migration under confinement

*D. discoideum* cells undergoing chemotaxis were recorded for 30 min at an interval of 1 min with a Zeiss LSM880 microscope using 514 nm excitation and a 20× objective. The resulting movies were imported into Fiji and analyzed using the MtrackJ plugin.

For confining fibroblasts, a 6-well glass bottom plate (4Dcell, Falcon/Mattek) was plasma cleaned and coated with 1 mg/mL pLL(20)-g [3.5]-PEG (2) (SuSoS) for 1 h at 37 °C. The plate was then washed twice with 1× PBS and $1 \times 10^5$ cells were seeded in DMEM culturing media. Cells were allowed to settle for 10 min before confining them with a static confiner (4Dcell, CSOW 610)[68] and imaged for 8 h with a Zeiss LSM880 microscope using a 20× objective (Supplementary Fig. 9d). For inhibiting myosin, 50 μM para-amino blebbistatin (Sigma, AA0200ZV) was added in 1 mL DMEM before confining the cells. The migration analysis was performed with TrackMate-StarDist algorithm[80].

### Immunoprecipitation, western blot and mass spectrometry

$1 \times 10^7$ cells were allowed to adhere overnight and washed with PBS or PB once, and then lysed using the M-PER Mammalian Protein Extraction Reagent (Thermo Fisher Scientific) containing a protease inhibitor cocktail (cOmplete ULTRA, mini, EDTA-free EASYpack, Roche) and phosphatase inhibitor (PhosSTOP, Roche). Lysates were centrifuged for 10 min at 4 °C and the supernatant was subjected to immunoprecipitation using μMACS GFP Isolation Kit (MACS Miltenyi Biotec, 130-091-125) according to the manufacturer protocol. Proteins eluted from the μMACS beads were then subjected to the SP3 protocol for sample preparation[81]. Desalting of these samples was accomplished using SDB-RPS tips (Affinisep). Nanoflow reversed-phase liquid chromatography was performed using a nanoElute 2 UHPLC system, online coupled to a timsTOF Pro2 mass spectrometer via a CaptiveSpray nano-electrospray source (all by Bruker Daltonics). Peptides were separated on a 25 cm × 75 μm fused silica capillary column (CoAnn Technologies), home-packed with 1.5 μm ReproSil Saphir 100 C18 beads (Dr. Maisch) using a linear 28 min - gradient from 4 to 27% buffer B (99.9% acetonitrile, 0.1% formic acid; buffer A: 0.1% formic acid) at a flow rate of 300 nl/min. The mass spectrometer was operated in DIA-PASEF mode, using the standard DIA PASEF short gradient MS-method. Signals covering a mass range from 100 to 1700 m/z and an ion mobility range from 1/K0 = 0.85 to 1.30 Vs/cm² using a ramp time in the dual TIMS analyzer of 100 ms were recorded. The collision energy was lowered as a function of increasing ion mobility from 59 eV at 1/K = 1.60 Vs/cm² to 20 eV at 1/K0 = 0.6 Vs/cm². Raw MS data were processed using DIA-NN (v. 1.8.2beta27 and v. 1.9) in the library free search mode, using heuristic protein inference on the genes level. Neural networks were used with single-pass mode, matching between runs was enabled, and as quantification strategy QuantUMS was applied. For the murine databases, entries were sourced from UniProt in January 2022; the *Dictyostelium* databases were from April 2024. Carbamidomethylation on cysteine residues was set as fixed modification for the search, while oxidation at methionine and acetylation of protein N-termini were set as variable modifications. Trypsin was defined as the digesting enzyme, allowing a maximum of two missed cleavages and a peptide length range of 6 or 7–35 amino acids. The maximum allowed mass deviation was 20 ppm for MS and 15 ppm for MS/MS scans. Charges above +5 were excluded. Protein groups were regarded as being unequivocally identified with a false discovery rate (FDR) of 1% for both the peptide and protein identifications. Downstream analysis was performed using the Perseus software (version 1.6.15.0). Common lab contaminants were removed from the dataset, protein intensity values as provided in the pg_matrix.tsv table were first log2 transformed, followed by filtering for at least three valid values in one of the experimental groups. Missing values were replaced by imputation using a left censored normal distribution (width = 0.3, down shift = 1.8) or by quantile regression imputation for left censored data (QRILC) from the R imputeLCMD package (sigma = 0.7). Significant differences between control and inhibitor-treated samples were determined using a Student's *t* test with S0 = 0.1 and permutation-based FDR calculations. Proteins with a *q*-value below 0.05 were reported as significant. All mass spectrometry proteomics data have been deposited to the ProteomeXchange Consortium via the PRIDE partner repository with the dataset identifiers[82] (ProteomeXchange accession: PXD062078). Protein lysates were analyzed through western blotting that was performed according to the standard wet transfer protocol. Data were quantified using densitometry with Fiji.

### Time-correlated single photon-counting fluorescence lifetime microscopy (TCSPC-FLIM)

Experiments were performed on live cells either seeded on FN-coated dishes for fibroblasts ($2 \times 10^4$ cells) or under agarose for *D. discoideum* ($1 \times 10^6$ cells) using an LSM880 confocal microscope (Zeiss) equipped with a 63× glycerin objective and a PicoQuant FLIM set up as described before[76]. The SymPhoTime64 (Picoquant) software was used to obtain the fluorescence lifetime using "n-Exponential Tailfit" function. The FRET efficiency ($E$) was calculated according to Eq. (2), where ($\tau_{DA}$) is the lifetime of the donor in presence of an acceptor and ($\tau_D$) is the experimental mean of the donor lifetime:

$$E = 1 - \frac{\tau_{DA}}{\tau_D} \tag{2}$$

### Statistics and reproducibility

Graphs were generated in OriginPro software. Error bars indicate the standard deviation (SD). Normally distributed data was analyzed by a

two-sample *t*-test (α = 0.05) to calculate the statistical significance. Box plots show the median and the 25th and the 75th percentile; the whiskers reach out to the last data point within 1.5× interquartile range corresponding to 2.7 SD. Statistical significance was determined with two sample Kolmogorov–Smirnov test (α = 0.05). Statistics for bar graphs was determined with a two sample Kolmogorov–Smirnov test or a One-way ANOVA (α = 0.05). Statistics for FRAP recovery curves were determined with a two-sample test for variance (α = 0.05). The *p*-values calculated are indicated in the figures as follows: ***$p < 0.001$; **$p < 0.01$; *$p < 0.05$; n.s. (not significant), $p \geq 0.05$. No statistical method was used to predetermine sample size. Data with insufficient signal intensity (photon counts) were manually excluded from the analysis. The experiments were not randomized and the investigators were not blinded to allocation during experiments and outcome assessment.

### Reporting summary
Further information on research design is available in the Nature Portfolio Reporting Summary linked to this article.

## Data availability
The data and statistical evaluations supporting the findings of this study are available within the article, the Supplementary Information, and the Source Data file. Mass spectrometry data generated in this study have been deposited at the ProteomeXchange Consortium via the PRIDE partner repository with accession PXD062078. Source data are provided with this paper.

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

## Acknowledgements

The authors would like to thank Prof. Dr. Jan Faix (MHH Hanover) for expert advice during initial stages of the project, Dr. Jürgen Eirich (University of Münster) for mass spectrometry sample preparation, Prof. Dr. Ralf Adams (MPI of Molecular Biomedicine) for support with mass spectrometry analysis, Thorsten König and Prof. Dr. Frank Rosenbauer (UKM Münster) for FACS sorting, and Lukas Windgaße for advice on cell tracking. S.R. is a member of CiM-IMPRS, the joint graduate school of the Cells-in-Motion Interfaculty Centre (University of Münster, Germany) and the International Max Planck Research School–Molecular Biomedicine (Münster, Germany). The research was supported by a grant of the Volkswagen Stiftung (Surviving under Pressure) and the German Research Foundation through the Collaborative Research Centre SFB1348 (project A12) to C.G.

## Author contributions

S.R. and L.E. performed experiments and analyzed the data. H.C.A.D. performed mass spectrometry analysis. A.Ch-G. designed cloning strategies and generated expression constructs. C.G. conceptualized and supervised the study, and wrote the paper with input from all authors.

## Funding

## Competing interests

The authors declare no competing interests.
