## [Transparent Peer Review file · Nature Communications]

Talin force coupling underlies eukaryotic cell-substrate adhesion

Corresponding Author: Professor Carsten Grashoff

Version 0:

Reviewer comments:

Reviewer #1

(Remarks to the Author)

Manuscript "Talin force coupling underlies eukaryotic cell-substrate adhesion" by Rangarajan, Grashoff and collaborators deals with the comparison of mammalian and Dictyostelium Talin's roles in cell substrate adhesion and cell migration processes.

The authors found that Dicto Talins A and B enable to engage any interaction with mammalian integrins (link and activation), any Rank-dependent focal adhesion recycling and related cell migration, as well as any mechanotransduction signalling upstream of Yap in mammalian cells, all behaviours characteristic of mammalian Talins^{1,2}.

They additionally find that, unless in the absence of integrins, they transmit at least 6-8pN forces to the actin cytoskeleton characteristic of mammals Talins^{1,2} through an interaction with an alternative plasma membrane protein SibA for TalA, in Dictyostelium migrating amoebae.

They finally find that the SibA-TalA Dictyostelium complex can promote substrate cytoskeleton mechanical coupling in mammalian cells, but not to spontaneous migration, except under a mechanically confined environment.

The authors conclude on an evolutionary conserved role of Talins in the mechanical coupling of the actin cytoskeleton with the substrate, with a role in the focal adhesion turn-over allowing cell migration and mechanotransduction signalling pathways having evolved late, in eukaryotic cells.

Comments:

This manuscript reports a very interesting piece of work, having discovered such shared role of Talins in the mechanical coupling of the actin cytoskeleton with the cell substrate in both a mammalian cell line and Dictyostelium cells, with the finding of SibA as the plasma membrane protein coupling to Dictyostelium TalA.

Based on these findings, evolutionary conclusions remain however a bit over-stated, and would require additional experiments, or more prudent formulations (including grammatical conditional ("may", "might") and terminologies like "possibly").

In addition, the impression after manuscript reading, is that the study remains somewhat incomplete with regard to the very interesting finding of a SibA-TalA Dictyostelium recues of migration in Tal^{1,2} deficient mammalian cells, under mechanical confinement only.

To progress on these two points, this reviewer would suggest the following:

1- Please detail why, precisely, attachment and migration of cells on ECM were crucial for evolution of animals, by giving concrete examples.

2- It would be very interesting to test if mechanical confinement stimulates migration in Dicto cells, like in Tln1,2-/- SibA-TalA mammalian cells. Does it enhance it? Can it be triggered on very soft substrates that would initially prevent it? Would it be the case, one could interestingly argue of a SibA-TalA mechanical stimulation of migration by either the substrate stiffness of the mechanical confinement, in Dicto.

3- Related to the precedent question, there should be a mechanically induced TalA dynamical turn-over in the Tln1,2-/- SibA-TalA mammalian cell submitted to mechanical confinement. As well as in Dicto confined cells (enhanced or triggered under low-stiffness conditions). This would be highly interesting to test, as it would show the existence of a mechanotransduction pathway upstream of the triggering of such turn-over leading to migration.

4- It would be interesting to test in the presence of SibA only if Tln1,2 interact with SibA in mammalian cells.

5- The SibA-TalA complex and Integrin-Tln1,2 complexes may either have diverged from a common ancestral complex with a common ability to link mechanically cell-substrate with the cytoskeleton; or being the fruit of convergent-evolution processes in both species. A well-chosen third species showing similar results would favour the evolutionary conserved interpretation as more probable. In the absence of it, it is difficult to conclude, and both the evolutionary conserved and convergent-evolution interpretations should be presented symmetrically.

6- The role of forces and mechanotransduction in evolution is an emerging field (involvement in multicellularity (DOI: 10.1126/science.adu0047) or primitive animals emergence (10.3389/fcell.2022.992371), strongly related to the study, should be better documented from the beginning of the introduction.

Reviewer #2

(Remarks to the Author)

The manuscript by Rangarajan et al. presents a compelling and well-executed study on the evolutionary origins and mechanistic conservation of talin as a force-transducing protein. The authors propose that talin emerged in unicellular organisms as a mechanical linker capable of transmitting piconewton-scale forces, and was later adapted in multicellular animals to mediate integrin-based adhesion and signaling, through specialized interactions with proteins such as paxillin, FAK, vinculin and KANK. The use of an interdisciplinary approach, to trace the functionality of talin from Dictyostelium discoideum to mammalian systems, offering insights into the evolutionary trajectory of mechanotransduction mechanisms.

The manuscript is well written and highly engaging. However, precisely because of its relevance and potential impact, it raises a number of questions and concerns that, in my view, warrant further clarification and revision by the authors. These relate to both the experimental design and the interpretation of key findings.

My detailed comments and suggestions are outlined below.

1. TalA and TalB are associated to different life-cycle stages of *D. discoideum* (single cell, vegetative vs. multicellular aggregate, mound, slug and fruiting body): what are the major differences in their rod structure that could justify their stage-specificity? Related to this, the authors should include the known and predicted interacting sites for actin, FAK, vinculin, paxillin, and KANK in Figure 1 a, in the Schematic depiction of the domain structure of human Tln1, and TalA and TalB from *D. discoideum*.

2. Do the authors think that the molecular forces generated between SibA– Discoidin and Tal A/B are the same as the ones between Itgβ-fibronectin and Tln 1/2.

3. The authors test the spreading capability of Tln1/2 -/- fibroblasts expressing Tln1 or TalA or TalB seeded on fibronectin. Are these experiments performed on glass? Wouldn't Agar be a more physiological substrate for *D. discoideum*? Can Tln1/2 -/- fibroblasts expressing TalA and TalB spread on agar? Do the authors think that the difference in rigidity between glass and solid agar could be a reasonable explanation for the difficulty in spreading of Tln1/2 -/- fibroblasts expressing TalA and TalB? In their beautiful Supplementary video 4 the authors indeed show the ability of TalA expressing *D. discoideum* to attach to agarose and migrate.

4. In the experiments of talin molecule turnover there should be a control with talin head only. Turnover of Talin has been reported to be regulated by talin head phosphorylation (<https://doi.org/10.1038/nrm2702>), thus the results that Tln1, chTalA and chTalB (all sharing the same head domain) show a similar turnover rate should not be surprising. That is important to show the talin head only results.

5. Related to the FA turnover, the authors conducted mass spec to determine interactome of Tln1, chTalA and chTalB. Was vinculin identified in the interactome of chTalA and chTalB? Please include the results about vinculin to support the interaction only suggested by immunostaining for vinculin. Additionally, the mass spec results should be incorporated as Supplementary tables or deposited in a Repository.

6. Methods for generation of Tln1/2 -/- fibroblasts and quantification by PCR or WB are missing. I think it would be important to know whether there is any residual Talin expression in these cells. Also, it is reported "Lack of talin expression in these cells results in a severe adhesion phenotype and an inability to form mechanically robust focal adhesions (FAs)" I would suggest including a representative image in Extended figure 1.

7. Extended figure 1, in Tln 1/2 *-/-* with TalA and with TalB, although much smaller, Itg 1 seems to form adhesive puncta. How are these adhesions supposedly activated? And what are they connected to intracellularly? Please include figures for Tln 1/2 *-/-* as comparison. Additionally small TalA and TalB aggregates are visible in perinuclear position in both Tln 1/2 *-/-* and Tln 1/2 *f/f*. Can you please comment on this?
8. It appears that a FRET control is missing—specifically, a construct containing only the Tln1 head fused to the TS, without the rod domain, similar to the Tln1-ctrl indicated in Extended Data figure 1d, but without the rod domain. Including this control would help clarify whether the observed FRET signal is indeed due to rod domain.
9. The authors suggest that the rod domains of the primeval talin proteins of *D. discoideum* can engage with vinculin based on FLIM-FRET experiments and immunostaining for vinculin at focal adhesions. While these observations are intriguing, the conclusion seems somewhat speculative in the absence of direct biochemical evidence. I would recommend supporting this claim with additional experiments, such as co-immunoprecipitation, to more robustly confirm the proposed interaction.
10. At the end of the chapter The primeval talin rod fails to induce signalling via paxillin, FAK and YAP, the authors report the following:
“Collectively, these data demonstrate that the primordial talin found in *D. discoideum* is capable of mediating a force-bearing mechanical linkage in cell-substrate adhesions.”
First of all, the authors should correct it indicating “the primordial talin ROD”; additionally the authors should corroborate their FLIM-FRET findings with single molecule magnetic tweezer experiments to effectively measure the functionality of TalA and TalB rod (without the Tln1 head) in bearing physiological forces. At last, if TalA and TalB rods are capable of bearing forces, it means they are supposedly in the mechanotransduction pathway: what are the potential anchoring partners?
11. In Figure 1i, chTalA shows two layers of adhesions and the most internal one is not vinculin associated. Can you explain what are these adhesions? Could this support the authors hypothesis of the participation of TalA in the recycling of adhesions? Are they associated to the matrix?
12. In Figure 4, data corresponding to all TalA constructs should be reported (as figures and numbers in the plots) at least as Extended data Figure.
13. In Figure 4e–f and Extended Data Figure 6, the authors show that *D. discoideum* development is not adversely affected by expression of the TalA constructs. However, I find the rationale for including this data somewhat unclear, particularly since the authors themselves state that the main functional distinction between TalA and TalB lies in their roles at different life stages, with TalA being primarily relevant during the single-cell stage. It would be helpful if the authors could clarify the motivation for this experiment and how it supports their overall conclusions.
14. The authors mention that “Mass spectrometry analysis of immunoprecipitates from TalA-/- amoebae expressing either the full-length TalA protein or the TalA head domain confirmed the complexation with actin-associated proteins, such as myosin-I heavy chain55 and α -actinin A56”. I am not sure I had access to the data so I could not verify: is there a direct interaction of talA and Actin? How about vinculin? Kank? Please note, the citation of Extended Data figure 6E I think it is misplaced.
15. At the end of the section Talin-A engages alternative transmembrane receptors for force coupling, the authors conclude that “These data demonstrate that the function of talin as a mechanical linker, connecting cell surface receptors to the actin cytoskeleton, is already established in amoebae and efficiently operates in the absence of integrin receptors.” Similarly, at the end of section The SibA-TalA linkage is sufficient to promote amoeboid migration in mammalian cells the authors states that “These data demonstrate that the primeval talin protein can promote mechanical coupling in mammalian cells when a suitable membrane adaptor such as SibA is provided, and they suggest that integrin-independent force coupling by talin can promote amoeboid migration of mammalian cells”
However, since SibA is described as an integrin β -like protein and TalA has been shown to mediate adhesion through SibA, the statement about the absence of integrin receptors seems somewhat misleading here and in the discussion as well. I would suggest rephrasing this conclusion to acknowledge the presence of integrin-like proteins such as SibA, or to clarify the authors reference to canonical integrins.
16. In figure 5, the authors show that truncation of the potential ABSs from the TalA rod or the absence of *ctnnA/vin-A* result in a loss of molecular tension across the TalA molecule. Have the authors performed similar experiments with their fibroblasts model?
17. The authors performed TalA force measurements in *SadA*-deficient *Dictyostelium* cells (*SadA*-/-); I think the authors should explain what is *SadA*: is it an adhesion receptor? How is it different from SibA?
18. What was the rationale of using fibroblasts? There are some cells, like GD25 cells that are fibroblasts derived from β -null mouse embryonic stem cells (E5-D3) that could be a great model to recapitulate the behaviour of Talin from *D. discoideum* in the absence of integrin. It would actually be a great model for their SibA and TalA reconstitution experiments. Furthermore, fibroblasts are known for secreting fibronectin while in *D. discoideum* another protein is present (Discoidin 1, [https://doi.org/10.1016/0092-8674\(84\)90462-8](https://doi.org/10.1016/0092-8674(84)90462-8)). Could this make a difference when comparing the two systems? What is the ECM protein used in the experiments with fibroblasts and with *D. discoideum*? Can you please discuss this?

19. It has been reported the existence of a nuclear Tln1 (<https://doi.org/10.1016/j.isci.2025.111745>). Could its role in the nucleus be related to talin's presence in unicellular organisms lacking canonical integrins? What is the amount of Talin in the nucleus in *D. Discoideum*? What is the amount of Talin at FA in *D. discoideum*? Representative figures and quantification should be included.

20. In general, for all fluorescence images in the manuscript, please provide grayscale for each color channel; additionally, if channels are merged, they should be accessible to color blind people and avoid rainbow colors. These are implementation guidelines recently reported in <https://pmc.ncbi.nlm.nih.gov/articles/PMC10922596/>.

Reviewer #3

(Remarks to the Author)

This is an interesting manuscript that seeks to delve into the evolution of focal adhesions and the authors find that talin's mechanical force coupling is conserved in organisms evolutionarily distant from metazoans. Using chimeras of talin mammalian head domains and Dicty rod domains, they find that the Dicty head domain does not with mammalian integrin, but the Dicty rod domains interact with mammalian focal adhesion, Vinculin. Using mass spec, they also identify distinct interactors with the mammalian talin rod domain (KANK1/2) and the Dicty rod domain (LPP). Using a tension sensor, the authors found that TalA exhibits "stretch" and can therefore transmit mechanical forces at structures that form on the ventral surface during Dicty cell migration. This mechanical transmission is dependent on the presence of SibA and SadA. The authors then expressed Dicty TalA in mammalian Tln1/2- fibroblasts and found that when TalA is expressed with SibA, the cells migrate faster by transitioning to an amoeboid-like migration. From these experiments, the authors conclude that the ancestral function of TalA is to promote mechanical coupling at focal adhesions, and TalA can do so in an integrin-independent manner. The combination of mammalian systems and Dictyostelium to uncover ancestral traits of focal adhesion components is a great strength of this manuscript. Dictyostelium focal adhesions are also hard to visualize, so quantifying TalinA tensions sensors in Dicty is also quite a feat. However, I have some concerns that limit my enthusiasm for this work. Below I outline these concerns:

1. A key point made in the manuscript is how Talin transmits mechanical forces in the absence of an integrin receptor. I agree that this is a very interesting question, but I am not convinced that Dictyostelium does not have integrin homologs. The Sib family of proteins, including SibA, has been hypothesized to be an integrin homolog based on sequence homology (Cornillon et al., *Eukaryotic Cell*, 2008; Cornillon et al. *EMBO Reports*, 2006; Lally et al., *Trends in Microbiol*, 1999). While I understand that a recent paper, which is cited by the authors, provides evidence that suggests that Sibs are not integrin homologs, the nature of this recent paper is to look broadly across the evolutionary tree of life and do not specifically focus on Dictyostelium discoideum as the previous papers do. I think the jury may still be out with respect to whether the Sibs are integrin homologs, but in the absence of a clear answer, it is not advisable to make a claim that Talin transmits force in an integrin-independent manner. The burden of proof is on the authors to show that SibA is not an integrin in Dictyostelium. Showing data that TalA/B does not interact with mammalian integrin in mammalian cells is not sufficient.

2. It is not clear that TalA is localizing to focal adhesion-like structures in Dictyostelium. The large dot TalA structures that form in Dictyostelium are peculiar. Are these hypothesized to be focal adhesion like structures in Dictyostelium? There are published reports on ventral surface punctae that form as the Dictyostelium cell protrudes and resolve as the cell retracts, which are reminiscent of focal adhesion in mammalian systems. Why does the TalA structure in this manuscript look so different than the previously described focal adhesion-like structures in Dictyostelium? Localizing this structure with other cell-substrate adhesions is advisable, particularly given the intriguing mass spec results which do not show complexation with other "canonical" focal adhesion proteins like Dictyostelium vinculin and paxillin. Without knowing what these structures are, it is difficult to make comparisons regarding talin function at focal adhesions in Dictyostelium versus mammalian cells. If the dot-like structure is indeed a focal adhesion-like structure, it is difficult to understand how one large structure in the back of the cell can promote cell migration.

3. What is the dynamic range of the talin biosensor? The negative control is performed with LatA, but is there a positive control to know the upper limit?

4. The hypothesis that mammalian cells transition to an integrin-independent amoeboid form of motility is not well-supported. It is clear that when SibA and TalA are expressed in Tln1/2- mammalian cells, the cells move faster, but there is no indication of the type of motility that they are using. Markers and pharmacological inhibitors for amoeboid migration would need to be used to support this statement.

5. The data representation at times is unclear, and the data are not consistently quantified throughout. In several figures, representative images or western blots are shown, and quantification across biological replicates is needed.

Minor Comments:

-The kymographs are difficult to interpret. What is 0 and 1? I could not find information in the legend or methods.

-It would be ideal to better highlight the previous work showing that Dictyostelium do not have FAK and Src homologs. At times, the manuscript is misleading regarding what previous work has already been done in Dictyostelium.

-What does the TalA tension sensor look like during directed cell migration? Does TalA still form the same large structure in the back of the cell? Does the structure still transmit force during directed cell migration when the cells are migrating at faster speeds?

Version 1:

Reviewer comments:

Reviewer #1

(Remarks to the Author)

The authors responded systematically to all of this reviewer's questions and significantly improved the manuscript.

Regarding point 2 of the discussion negative results of the manuscript, the already confined mechanical status of Dicty cells in stiffness-dependent assays does indeed make it difficult to interpret the data, particularly if excessive confinement triggers cell death (which should be verified by labeling cell death markers) or any other process leading to rounding. This would indeed warrant further research to be published and can be omitted at this stage.

Regarding point 4 of the discussion negative results of the manuscript, this challenging question would certainly warrant further research, which would take a long time, to be conducted as part of future work.

Regarding point 3 of the discussion, the absence of obvious change in the spatial distribution of Tal4 is an interesting result, indicating the absence of detectable membrane Tal4 recycling. Frap experiments at membranes would certainly provide a definitive answer to this question, allowing the potential existence of local recycling helping migration, modifying not a steady state overall distribution at the cellular level.

If such an experiment is feasible in the context of the configuration of the present experimental device, it would certainly be nice trying it before publication.

Reviewer #2

(Remarks to the Author)

I would like to thank the authors for their thorough and thoughtful responses to my previous comments. I appreciate the significant effort that went into performing the additional experiments, expanding the datasets, and revising the text and figures. The authors have addressed all of my concerns in a clear and comprehensive manner.

I believe the new data and analyses strengthen the manuscript and further support the main conclusions. The presentation is now clear, balanced, and well-structured. I am satisfied with the revisions and have no remaining concerns.

In my opinion, the manuscript is now suitable for publication in Nature Communications.

Reviewer #3

(Remarks to the Author)

This resubmission contains additional data shoring up some of the take-home messages in the original manuscript, however, I'm afraid that some of my main concerns remain.

1. A key point in the manuscript and highlighted in the abstract is that Talin transmits mechanical forces in the absence of an integrin receptor ("we uncover a conserved role for talin in transmitting pN-scale forces into cells, even in unicellular organisms lacking integrin receptors.") This is a provocative statement, but one that I still do not feel is justified by the data. The authors have revised the text to convey the complexity of whether Dicty has a bona fide integrin homologue, and I understand the authors' desire to write such a compelling statement. The talin-centric cell biology is top notch, with beautiful characterization and quantification, but I am not convinced that talin being able to transmit mechanical forces without an integrin is a supported main take-home message of the paper, which will of course limit my overall enthusiasm for the work.

2. The authors include new data showing PaxB localization to the posterior TalA structures in migrating cells, but there is no quantification to facilitate the interpretation. What percentage of TalA-containing structures also localize PaxB.

3. Thank you for including additional experiments to better interpret the talin biosensor dynamic range.

4. I'm still having a tough time understanding the Tln1/2- mammalian experiment expressing SibA and/or TalA. This experiment was performed to test whether "the original, mechanical function of talin is sufficient to modulate processes". If the Tln1/2- cells expressing SibA/TalA are indeed undergoing bleb-based myoII-dependent migration (supported by the addition of the new blebbistatin experiments, which are nice), and TalA does not appear to localize in any punctate manner that interacts with ECM, how do these results support a mechanical function or coupling for TalA? Wouldn't the results of this experiment just suggest that TalA can promote bleb-based migration when SibA is present, but not due to adhesion to ECM and force transmission through TalA?

5. Thank you for including the requested quantification.

Response to Reviewer #1

This manuscript reports a very interesting piece of work, having discovered such shared role of Talins in the mechanical coupling of the actin cytoskeleton with the cell substrate in both a mammalian cell line and Dictyostelium cells, with the finding of SibA as the plasma membrane protein coupling to Dictyostelium TalA.

Based on these findings, evolutionary conclusions remain however a bit over-stated, and would require additional experiments, or more prudent formulations (including grammatical conditional (“may”, “might”) and terminologies like “possibly”).

In addition, the impression after manuscript reading, is that the study remains somewhat uncomplete with regard to the very interesting finding of a SibA-TalA Dictyostelium recues of migration in Tal1,2 deficient mammalian cells, under mechanical confinement only.

Response: We thank the reviewer for evaluating our manuscript and for the encouraging and constructive remarks. As suggested, we now tried to use more prudent formulations. In addition, we have addressed the points raised by the reviewer, performed additional experiments, generated new figures, and adjusting the text. Together with the additional changes requested by the other reviewers, we think that this has further improved the manuscript and we hope that the reviewer can now fully support the publication of our study.

To progress on these two points, this reviewer would suggest the following:

1- Please detail why, precisely, attachment and migration of cells on ECM were crucial for evolution of animals, by giving concrete examples.

Response: We have revised the introduction to provide more detailed information on the role of cell-extracellular matrix (ECM) interaction in animals. We emphasize that gastrulation, a defining feature of animal development, depends on cell-ECM interaction. We also refer to the formation of epithelia and the migration of neural crest cells, both of which require resilient cell-ECM linkages. We believe this information is useful, and together with the newly included references, it provides readers with a better introduction to the role of cell-ECM interactions in animal evolution. Thank you for this suggestion.

2- It would be very interesting to test if mechanical confinement stimulates migration in Dicto cells, like in Tln1,2-/- SibA-TalA mammalian cells. Does it enhance it? Can it be triggered on very soft substrates that would initially prevent it? Would it be the case, one could interestingly argue of a SibA-TalA mechanical stimulation of migration by either the substrate stiffness of the mechanical confinement, in Dicto.

Response: Please note that the *Dictyostelium* migration assays were performed in the so-called under agar assay (now Extended Data Figure 9a), and the cells are already confined by the overlying agar substrate. We still performed the suggested experiments and confined *Dictyostelium* cells as we did for the Tln1,2-/- SibA-TalA fibroblasts using the PDMS micropillar assay (Extended Data Figure 9d). We tested

confinement heights of 1 μm , 3 μm and 5 μm , but under all tested conditions *Dictyostelium* cells quickly rounded up and did not move at all (in fact, we believe the cells died upon confinement). We prepared an exemplary figure for the reviewer to illustrate the effect (please see below, Fig. 1). We conclude that the suggested assay is unsuitable to study confinement effects in our system and would like to refrain from showing the data.

Fig. 1: Confining *Dictyostelium* cells with glass slides at heights of 1, 3 and 5 μm abrogates cell migration in our system. Arrows are pointing towards the same cells that did not move upon confinement.

3- Related to the precedent question, there should be a mechanically induced TalA dynamical turn-over in the Tln1,2-/- SibA-TalA mammalian cell submitted to mechanical confinement. As well as in Dicto confined cells (enhanced or triggered under low-stiffness conditions). This would be highly interesting to test, as it would show the existence of a mechanotransduction pathway upstream of the triggering of such turn-over leading to migration.

Response: To test for a potential induction of Talin-A turnover upon confinement, we used Tln1/2-deficient cells that had been reconstituted with fluorescently tagged Talin-A and SibA. Live cell fluorescence imaging revealed a homogeneous distribution of Talin in cells under confinement. No obvious differences in the subcellular localization of talin-A were observed between static and migratory cells. Images of this experiment are now shown in the new Extended Data Fig. 8e.

4- It would be interesting to test in the presence of SibA only if Tln1,2 interact with SibA in mammalian cells.

Response: This is a challenging experiment as it requires cells expressing SibA and Talin, in the absence of integrin receptors, which would certainly interfere with a potential SibA-talin-1 interaction. We therefore used cells that are genetically depleted of all integrin receptors by deletion of $\beta 1$, $\beta 2$, αv and $\beta 7$ integrin and expressed SibA and talin-1. We then performed co-immunoprecipitation of talin and checked for SibA but did not find a significant enrichment of SibA in the IP eluate (see below, Fig. 2). While this indicates that SibA does not strongly engage talin-1, we feel that additional experiments would be required to entirely rule out an interaction. We would therefore prefer to not include this negative data set in the manuscript.

Fig. 2: Tln1 and SibA or only SibA alone were co-expressed in integrin-deficient cells. While talin-1 (Tln1) was successfully immunoprecipitated from lysates of Tln1/SibA cells, SibA was not significantly enriched in these samples indicating only weak or non-existing interactions.

5- The SibA-TalA complex and Integrin-Tln1,2 complexes may either have diverged from a common ancestral complex with a common ability to link mechanically cell-substrate with the cytoskeleton; or being the fruit of convergent-evolution processes in both species. A well-chosen third species showing similar results would favour the evolutionary conserved interpretation as more probable. In the absence of it, it is difficult to conclude, and both the evolutionary conserved and convergent-evolution interpretations should be presented symmetrically.

Response: We agree with the reviewer that experiments in a third (and probably even more species) are needed to fully clarify the evolutionary mechanisms. However, establishing a new model system and performing reproducible experiments is hardly possible in the course of this revision, and we hope the reviewer will understand. We therefore followed the reviewer's advice and discuss this issue. Given the available data on the phylogenetics of talin (Sebe-Pedros et al. *PNAS*, 2010; Fig. S5), we based our interpretation on the evolutionary conservation model, but we also point out that experiments in additional species are needed to establish the underlying, evolutionary mechanisms.

6- The role of forces and mechanotransduction in evolution is an emerging field (involvement in multicellularity (DOI: [10.1126/science.adu0047](https://doi.org/10.1126/science.adu0047)) or primitive animals emergence ([10.3389/fcell.2022.992371](https://doi.org/10.3389/fcell.2022.992371)), strongly related to the study, should be better documented from the beginning of the introduction.

Response: Thank you for this excellent suggestion. We are more than happy to include these very interesting papers and now cite the suggested work in the first paragraph of the manuscript.

Response to Reviewer #2

The manuscript by Rangarajan et al. presents a compelling and well-executed study on the evolutionary origins and mechanistic conservation of talin as a force-transducing protein. The authors propose that talin emerged in unicellular organisms as a mechanical linker capable of transmitting piconewton-scale forces, and was later adapted in multicellular animals to mediate integrin-based adhesion and signaling, through specialized interactions with proteins such as paxillin, FAK, vinculin and KANK. The use of an interdisciplinary approach, to trace the functionality of talin from Dictyostelium discoideum to mammalian systems, offering insights into the evolutionary trajectory of mechanotransduction mechanisms.

The manuscript is well written and highly engaging. However, precisely because of its relevance and potential impact, it raises a number of questions and concerns that, in my view, warrant further clarification and revision by the authors. These relate to both the experimental design and the interpretation of key findings.

Response: We thank the reviewer for carefully evaluating our manuscript, the positive remarks and the constructive suggestions. In response to these comments, we have performed additional experiments, assembled new sets of data, and adjusted the main text and figures. We think that these changes, together with a number of additional experiments requested by the other reviewers, significantly improved the manuscript further. We hope the reviewer can now support the publication of our study.

My detailed comments and suggestions are outlined below.

1. TalA and TalB are associated to different life-cycle stages of D. discoideum (single cell, vegetative vs. multicellular aggregate, mound, slug and fruiting body): what are the major differences in their rod structure that could justify their stage-specificity? Related to this, the authors should include the known and predicted interacting sites for actin, FAK, vinculin, paxillin, and KANK in Figure 1 a, in the Schematic depiction of the domain structure of human Tln1, and TalA and TalB from D. discoideum.

Response: We thank the reviewer for this suggestion and prepared a figure, in which we provide more detailed information on Talin-1, Talin-A and Talin-B. Due to space limitations in the main figure, we would like to show this information in the Extended Data Fig. 1a. This new Figure now provides information on the overall domain structure of Talin-1, Talin-A, and Talin-B and sequence similarities. We also indicate areas where known (and here identified) proteins are thought to interact.

2. Do the authors think that the molecular forces generated between SibA– Discoidin and Tal A/B are the same as the ones between Itgβ-fibronectin and Tln 1/2.

Response: This is a good question, but a very difficult one to answer experimentally. In mammalian cells, forces acting on intracellular proteins such as talin and vinculin are in the low pN range (Austen et al., *NCB*, 2015; Kanoldt et al., *Nat Comm*, 2020). However, extracellular forces at the integrin-fibronectin interface can be as high as 40 pN (Wang et al., *Science*, 2013), with some studies suggesting forces of over

100 pN (Galior et al, *Nano Letters*, 2015). Collectively, these findings indicate that force measurements across intracellular proteins cannot easily be related to extracellular forces. To answer this question properly, single-molecule atomic force microscopy (AFM) measurements of purified SibA–Discoidin linkages would be required, which is clearly beyond the scope of this study. Nevertheless, we have included a sentence in the discussion emphasizing the need to study extracellular forces and potential catch-bond dynamics in *Dictyostelium* linkages.

3. The authors test the spreading capability of Tln1/2 -/- fibroblasts expressing Tln1 or TalA or TalB seeded on fibronectin. Are these experiments performed on glass? Wouldn't Agar be a more physiological substrate for D. discoideum? Can Tln1/2 -/- fibroblasts expressing TalA and TalB spread on agar? Do the authors think that the difference in rigidity between glass and solid agar could be a reasonable explanation for the difficulty in spreading of Tln1/2 -/- fibroblasts expressing TalA and TalB? In their beautiful Supplementary video 4 the authors indeed show the ability of TalA expressing D. discoideum to attach to agarose and migrate.

Response: As described in the materials and methods and shown in Extended Data Fig. 9, experiments for both fibroblasts and amoeba are performed on ibi-treat μ -dishes, a surface with a stiffness similar to glass. Please note that the *Dictyostelium* cell in Supplementary Video 4 is also adhering to the ibi-treat μ -dish, but is overlaid with agar to facilitate chemotactic migration (see Extended Data Fig. 9). Since slime molds naturally adhere to a wide range of rather stiff 2D surfaces, we would not consider this an unphysiological substrate. However, a glass surface is of course a rather unphysiological setting for fibroblasts. We therefore followed the suggestion of the reviewer and performed experiments on hydrogels covering a physiological stiffness range and imaged cells expressing TalA and TalB but also chimeric TalA/B constructs. In all cases, experiments were consistent with our findings on glass: Tln1/2 -/- cells expressing either TalA or TalB were not able to spread, even on soft substrates, consistent with the notion that integrin activation is impaired. Chimeric cells displayed a phenotype similar to the one described on glass, with multiple rows of FAs while control cells were forming distinct FAs. The data, including the quantification of cell spreading on soft substrates, are now shown in the Extended Data Fig. 2d, e and in Extended Data Fig. 8b-d.

4. In the experiments of talin molecule turnover there should be a control with talin head only. Turnover of Talin has been reported to be regulated by talin head phosphorylation (<https://doi.org/10.1038/nrm2702>), thus the results that Tln1, chTalA and chTalB (all sharing the same head domain) show a similar turnover rate should not be surprising. That is important to show the talin head only results.

Response: We performed the suggested experiments by expressing the talin-head domain tagged with YPet in Tln1/2 -/- fibroblasts and performing FRAP experiments. The resulting data demonstrate that the initial recovery of the talin-head only protein is significantly faster than that of the full-length talin constructs, showing that the head domain alone does not determine talin's turnover dynamics and that the rod domain is required to anchor (and thereby immobilize) the protein in the adhesion structure. These data are now shown in Extended Data Fig. 3b.

5. Related to the FA turnover, the authors conducted mass spec to determine interactome of Tln1, chTala and chTalB. Was vinculin identified in the interactome of chTala and chTalB? Please include the results about vinculin to support the interaction only suggested by immunostaining for vinculin. Additionally, the mass spec results should be incorporated as Supplementary tables or deposited in a Repository.

Response: Yes, we detected vinculin in the Mass-Spec experiments (and a significant enrichment in chTala samples) and now show the respective data in the main figure (Fig. 2e). As indicated in the methods section, all mass spectrometry proteomics data will be deposited to the ProteomeXchange Consortium (<http://proteomecentral.proteomexchange.org>).

6. Methods for generation of Tn1/2 -/- fibroblasts and quantification by PCR or WB are missing. I think it would be important to know whether there is any residual Talin expression in these cells. Also, it is reported "Lack of talin expression in these cells results in a severe adhesion phenotype and an inability to form mechanically robust focal adhesions (FAs)" I would suggest including a representative image in Extended figure 1.

Response: Please note that the generation and characterization of the of Tn1/2 -/- fibroblasts was described in detail before (Austen et al, *NCB*, 2015; Theodosiou et al, *eLife*, 2016); these cells were extensively used by us and others (e.g., Ringer et al, *Nat Meth*, 2017; Azizi et al, *Sci Rep*, 2021; Sadhanasatish et al, *Sci Adv*, 2023). The original studies are cited in the manuscript and demonstrate the absence of talin-1 and talin-2 expression in these cells. Please also note that the Tn1/2 -/- cells expressing TalA or TalB essentially phenocopy Tn1/2 -/- cells under normal culture conditions (Fig. 1b). In addition, an image of the Tn1/2 -/- cells are now shown in Extended Data Fig. 1c.

7. Extended figure 1, in Tln 1/2 -/- with TalA and with TalB, although much smaller, Itgβ1 seems to form adhesive puncta. How are these adhesions supposedly activated? And what are they connected to intracellularly? Please include figures for Tln 1/2 -/- as comparison. Additionally small TalA and TalB aggregates are visible in perinuclear position in both Tln 1/2 -/- and Tln 1/2 f/f. Can you please comment on this?

Response: We do not think that the integrin β1 signal indicates 'adhesive' puncta. We observe similar signals in cells Tln1/2 -/- cells (now shown in Extended Data Fig. 1c), which are deficient in integrin activation (Theodosiou et al, *eLife*, 2016). This indicates that the dots are a background staining of the antibody and we point this out in the main text.

The TalA and TalB aggregates may have been a consequence of the TagBFP fusion, which occasionally induces small accumulations at the perinuclear region. We therefore generated new constructs, in which the talin proteins were fused with YPet and expressed them in Tln 1/2 -/- and Tln 1/2 f/f cells. As seen from the new images, we observe a largely homogenous distribution without dot-like structures. These new images are now shown in Fig. 1b and Extended Data Fig. 1b, d, e.

8. It appears that a FRET control is missing—specifically, a construct containing only the Tln1 head fused to the TS, without the rod domain, similar to the Tln1-ctrl indicated in Extended Data figure 1d, but without

the rod domain. Including this control would help clarify whether the observed FRET signal is indeed due to rod domain.

Response: We appreciate the comment but would like to point out that it is debatable whether this control is indeed more suitable, as deletion of the rod-domain affects the subcellular localization and also does not properly rescue the talin-knockout phenotype. However, as suggested, we generated the construct, expressed it in *Tln1/2* *-/-* cells and performed the requested experiments. FRET efficiencies in cells expressing this control are significantly higher than observed in full length constructs, confirming that the effect that we describe is indeed mediated by the rod-domain. These new data are now shown in Extended Data Fig. 2g.

9. The authors suggest that the rod domains of the primeval talin proteins of D. discoideum can engage with vinculin based on FLIM-FRET experiments and immunostaining for vinculin at focal adhesions. While these observations are intriguing, the conclusion seems somewhat speculative in the absence of direct biochemical evidence. I would recommend supporting this claim with additional experiments, such as co-immunoprecipitation, to more robustly confirm the proposed interaction.

Response: We agree with the reviewer that our findings are not evidence for direct binding of vinculin to talin in *Dictyostelium*. In fact, we did not find vinculin in the TalA interactome even though other, known TalA binding partners were identified (MyoI). To be clearer on this point, we adjusted the text in two places of the manuscript to emphasize this finding. We point out that effects may be indirect or that the direct interaction between talin-A and VinA may be transient and stabilized through mechanical tension.

10. At the end of the chapter The primeval talin rod fails to induce signalling via paxillin, FAK and YAP, the authors report the following: "Collectively, these data demonstrate that the primordial talin found in D. discoideum is capable of mediating a force-bearing mechanical linkage in cell-substrate adhesions. First of all, the authors should correct it indicating "the primordial talin ROD"; additionally the authors should corroborate their FLIM-FRET findings with single molecule magnetic tweezer experiments to effectively measure the functionality of TalA and TalB rod (without the Tln1 head) in bearing physiological forces. At last, if TalA and TalB rods are capable of bearing forces, it means they are supposedly in the mechanotransduction pathway: what are the potential anchoring partners?"

Response: We slightly modified the last sentence and now conclude that our data suggest that the primordial talin is capable of mediating force bearing linkages. While we appreciate the reviewer's second suggestion, we respectfully disagree that magnetic tweezers experiments are suitable for corroborating our findings. The ability of a protein or protein fragment to bear mechanical force in a tweezer experiment does not demonstrate that it does so in the context of a living cell and is part of a mechanotransduction pathway. Any protein can be purified and subjected to mechanical force using a magnetic tweezer or an optical trap. As long as forces are in the physiological range, it will bear the applied force but this is not proof for the protein to bear physiological forces. We hope the reviewer will understand that we did not perform this particular experiment.

Our data suggest that the N-terminal anchoring partner is SibA at the plasma membrane and the actomyosin network at the C-terminal rod domain, as described in the main text.

11. In Figure 1i, chTalA shows two layers of adhesions and the most internal one is not vinculin associated. Can you explain what are these adhesions? Could this support the authors hypothesis of the participation of TalA in the recycling of adhesions? Are they associated to the matrix?

Response: We are hesitant to conclude that these dot-like structures are of physiological relevance, because we see very similar signals outside cells indicating unspecific binding of the antibody (see Fig. 1i, chTalB, Vcl). To avoid confusion, we replaced the images and now show data in which the unspecific signals are less prominent.

12. In Figure 4, data corresponding to all TalA constructs should be reported (as figures and numbers in the plots) at least as Extended data Figure.

Response: We now show the data of all TalA constructs in the Extended Data Fig. 6a. The reader can now appreciate that all TalA construct rescue the cell adhesion and phagocytosis defect, albeit to slightly different extent.

13. In Figure 4e–f and Extended Data Figure 6, the authors show that D. discoideum development is not adversely affected by expression of the TalA constructs. However, I find the rationale for including this data somewhat unclear, particularly since the authors themselves state that the main functional distinction between TalA and TalB lies in their roles at different life stages, with TalA being primarily relevant during the single-cell stage. It would be helpful if the authors could clarify the motivation for this experiment and how it supports their overall conclusions.

Response: We performed these experiments because the expression of the talin constructs used to rescue the talin-A-deficient amoeba is driven by an exogenous promoter, resulting in constitutive expression. It was therefore possible that ongoing expression during developmental stages, in which talin-A expression is typically reduced (Tsujioka et al, *Eukaryotic Cell*, 2008), might have a dominant negative effect. Our experiments demonstrate that this is not the case, and it seemed important to document this finding. We have adjusted the main text to better explain the rationale behind the experiment.

14. The authors mention that "Mass spectrometry analysis of immunoprecipitates from TalA-/- amoebae expressing either the full-length TalA protein or the TalA head domain confirmed the complexation with actin-associated proteins, such as myosin-I heavy chain55 and α -actinin A56". I am not sure I had access to the data so I could not verify: is there a direct interaction of talA and Actin? How about vinculin? Kank? Please note, the citation of Extended Data figure 6E I think it is misplaced.

Response: The data are shown in Extended Data Fig. 7a. We see the enrichment of actin and myosin isoforms as well as actin-associated proteins like α -actinin. As mentioned above, vinculin was not

identified and KANK is also not present. The full data set will be uploaded to the ProteomeXchange Consortium (<http://proteomecentral.proteomexchange.org>).

15. At the end of the section Talin-A engages alternative transmembrane receptors for force coupling, the authors conclude that "These data demonstrate that the function of talin as a mechanical linker, connecting cell surface receptors to the actin cytoskeleton, is already established in amoebae and efficiently operates in the absence of integrin receptors." Similarly, at the end of section The SibA-TalA linkage is sufficient to promote amoeboid migration in mammalian cells the authors states that "These data demonstrate that the primeval talin protein can promote mechanical coupling in mammalian cells when a suitable membrane adaptor such as SibA is provided, and they suggest that integrin-independent force coupling by talin can promote amoeboid migration of mammalian cells" However, since SibA is described as an integrin β -like protein and TalA has been shown to mediate adhesion through SibA, the statement about the absence of integrin receptors seems somewhat misleading here and in the discussion as well. I would suggest rephrasing this conclusion to acknowledge the presence of integrin-like proteins such as SibA, or to clarify the authors reference to canonical integrins.

Response: We understand that the issue needs to be discussed in more detail and we adjusted the text accordingly. We point out that SibA, while sharing features with integrins, such as the NPXY motif in the cytoplasmic region and the VWA domain in the extracellular region, is not an integrin receptor. All known integrin receptors are constitutive heterodimers consisting of an α - and a β -subunit. The ectodomain of both α - and β -integrins, which together mediate ligand binding and are conformationally regulated, differ significantly from the extracellular part of SibA, which lacks many of the characteristic domains found in integrin receptors. Please also note that other transmembrane proteins carry NPXY motifs (e.g., certain lectins) or VWA domains (e.g., cation channels). The authors of the original SibA study (Cornillon et al, *EMBO Reports*, 2006) acknowledged these differences, speaking of a SibA being a functional equivalent of integrins. In the absence of phylogenetic data directly connecting SibA with integrin receptors, we would like to follow this line of argument and point out that this 'functional equivalent' is utilized by talin to facilitate force coupling in the absence of heterodimeric integrin receptors. We adjusted the text accordingly.

16. In figure 5, the authors show that truncation of the potential ABSs from the TalA rod or the absence of ctnnA/vin-A result in a loss of molecular tension across the TalA molecule. Have the authors performed similar experiments with their fibroblasts model?

Response: Yes, we have performed these experiments in earlier studies (Austen et al, *NCB*, 2015; Ringer et al, *Nat Meth*, 2017). In particular, the study by Ringer et al shows that the actin binding sites determine the amount of force across talin. Similar findings have been obtained by another group (Kumar et al, *JCB*, 2016).

17. The authors performed TalA force measurements in SadA-deficient Dictyostelium cells (SadA-/-); I think the authors should explain what is SadA: is it an adhesion receptor? How is it different from SibA?

Response: We have included a sentence on SadA in the main text and also cite the original reference (Fey et al., JCB, 2002). Along with the already mentioned study (Froquet et al, *MBoC*, 2012), this gives readers a clearer idea of what is currently known about SadA.

*18. What was the rationale of using fibroblasts? There are some cells, like GD25 cells that are fibroblasts derived from β -null mouse embryonic stem cells (E5-D3) that could be a great model to recapitulate the behaviour of Talin from *D. discoideum* in the absence of integrin. It would actually be a great model for their SibA and TalA reconstitution experiments. Furthermore, fibroblasts are known for secreting fibronectin while in *D. discoideum* another protein is present (Discoidin 1, [https://doi.org/10.1016/0092-8674\(84\)90462-8](https://doi.org/10.1016/0092-8674(84)90462-8)). Could this make a difference when comparing the two systems? What is the ECM protein used in the experiments with fibroblasts and with *D. discoideum*? Can you please discuss this?*

Response: GD25 cells lack the expression of β 1 integrin receptors but they express other integrins such as α v β 3 and α v β 5 (Wennerberg et al, JCB, 1996). As talin readily binds β 3 integrins, the system is less powerful than expected by the reviewer, at least for our studies. In addition, these cells express endogenous talin-1 which would complicate TalA reconstitution experiments. Therefore, GD25 cells do not provide significant advantages over the here used system. As indicated in the manuscript, fibroblast experiments were performed on fibronectin-coated surfaces, while *Dictyostelium* was cultured on uncoated surfaces. We do not see how using fibronectin for *Dictyostelium* experiments would yield additional insights, because amoeba lack specific fibronectin receptors.

*19. It has been reported the existence of a nuclear tln1 (<https://doi.org/10.1016/j.isci.2025.111745>). Could its role in the nucleus be related to talin's presence in unicellular organisms lacking canonical integrins? What is the amount of Talin in the nucleus in *D. Discoideum*? What is the amount of Talin at FA in *D. discoideum*? Representative figures and quantification should be included.*

Response: Please excuse us for saying so, but we do not quite understand how this point is related to our study. While this is certainly an interesting question, we do not see how answering it helps understanding the here described effects. Our live cell imaging and immunostaining of talin-A-YPet expressing amoeba indicates that the protein is not strongly enriched in the nucleus. We also did not detect nucleus-specific proteins in the talin-A interactome. Based on these observations, we would conclude that the amount of talin in the nucleus is at least low. That being said, it will be interesting to explore a nuclear function of talin in the nucleus, but this would be an entirely different study. The amount of talin in focal adhesion cannot be quantified because *Dictyostelium* does not form these types of structures under the here used conditions (please see Supplementary Movie 4).

20. In general, for all fluorescence images in the manuscript, please provide grayscale for each color channel; additionally, if channels are merged, they should be accessible to color blind people and avoid rainbow colors. These are implementation guidelines recently reported in <https://pmc.ncbi.nlm.nih.gov/articles/PMC10922596/>.

Response: We fully agree with the reviewer that images should use colors that are suitable for color-blind readers. We therefore followed the Nature research figure guide (

[guide.nature.com/figures/preparing-figures-our-specifications/#colour-space](https://www.nature.com/figures/preparing-figures-our-specifications/#colour-space)). Consistent with these guidelines, we made sure to use green and magenta wherever possible (instead of green and red, or green and yellow) in our fluorescence images. In three-color images, we combine green, magenta and blue, as suggested in the Nature research figure guide. We hope that we have corrected all potential problems, so that color-blind readers can fully appreciate the data.

Response to Reviewer #3

This is an interesting manuscript that seeks to delve into the evolution of focal adhesions and the authors find that talin's mechanical force coupling is conserved in organisms evolutionarily distant from metazoans. Using chimeras of talin mammalian head domains and Dicty rod domains, they find that the Dicty head domain does not with mammalian integrin, but the Dicty rod domains interact with mammalian focal adhesion, Vinculin. Using mass spec, they also identify distinct interactors with the mammalian talin rod domain (KANK1/2) and the Dicty rod domain (LPP). Using a tension sensor, the authors found that TalA exhibits "stretch" and can therefore transmit mechanical forces at structures that form on the ventral surface during Dicty cell migration. This mechanical transmission is dependent on the presence of SibA and SadA. The authors then expressed Dicty TalA in mammalian Tln1/2- fibroblasts and found that when TalA is expressed with SibA, the cells migrate faster by transitioning to an amoeboid-like migration. From these experiments, the authors conclude that the ancestral function of TalA is to promote mechanical coupling at focal adhesions, and TalA can do so in an integrin-independent manner. The combination of mammalian systems and Dictyostelium to uncover ancestral traits of focal adhesion components is a great strength of this manuscript. Dictyostelium focal adhesions are also hard to visualize, so quantifying TalinA tensions sensors in Dicty is also quite a feat. However, I have some concerns that limit my enthusiasm for this work. Below I outline these concerns:

Response: We thank the reviewer for carefully evaluating our manuscript and making very constructive and informed suggestions. In response to these remarks and comments from the other reviewers, we have adjusted the text, included new data and prepared additional figures. We think that these changes further improved the manuscript and we hope the reviewer can now fully support the publication of our data sets.

1. A key point made in the manuscript is how Talin transmits mechanical forces in the absence of an integrin receptor. I agree that this is a very interesting question, but I am not convinced that Dictyostelium does not have integrin homologs. The Sib family of proteins, including SibA, has been hypothesized to be an integrin homolog based on sequence homology (Cornillon et al., Eukaryotic Cell, 2008; Cornillon et al, EMBO Reports, 2006; Lally et al., Trends in Microbiol, 1999). While I understand that a recent paper, which is cited by the authors, provides evidence that suggests that Sibs are not integrin homologs, the nature of this recent paper is to look broadly across the evolutionary tree of life and do not specifically focus on Dictyostelium discoideum as the previous papers do. I think the jury may still be out with respect to whether the Sibs are integrin homologs, but in the absence of a clear answer, it is not advisable to make a claim that Talin transmits force in an integrin-independent manner. The burden of proof is on the authors to show that SibA is not an integrin in Dictyostelium. Showing data that TalA/B does not interact with mammalian integrin in mammalian cells is not sufficient.

Response: We agree with the reviewer that this issue warrants a more detailed discussion in the manuscript, and we have amended the main text accordingly. In our opinion, it is difficult to argue that SibA is an integrin. All known integrin receptors are constitutive heterodimers consisting of an α - and a β -subunit with highly specialized functions. The binding affinity of integrins is highly specific to particular heterodimers and regulated through conformational changes involving protein domains, which are absent in SibA. The authors of the original SibA study (Cornillon et al, *EMBO Reports*, 2006) acknowledged these

differences, speaking of SibA being a ‘functional equivalent’ of integrins. In the absence of phylogenetic data directly connecting SibA with integrin receptors, we would like to follow this line of argument and point out that this ‘functional equivalent’ is utilized by talin to facilitate force coupling in the absence of heterodimeric integrin receptors that we know from metazoans. We tried to clarify this point in the main text and the discussion, and we thank the reviewer for pointing out that this needed to be described better.

2. It is not clear that TalA is localizing to focal adhesion-like structures in Dictyostelium. The large dot TalA structures that form in Dictyostelium are peculiar. Are these hypothesized to be focal adhesion like structures in Dictyostelium? There are published reports on ventral surface punctae that form as the Dictyostelium cell protrudes and resolve as the cell retracts, which are reminiscent of focal adhesion in mammalian systems. Why does the TalA structure in this manuscript look so different than the previously described focal adhesion-like structures in Dictyostelium? Localizing this structure with other cell-substrate adhesions is advisable, particularly given the intriguing mass spec results which do not show complexation with other “canonical” focal adhesion proteins like Dictyostelium vinculin and paxillin. Without knowing what these structures are, it is difficult to make comparisons regarding talin function at focal adhesions in Dictyostelium versus mammalian cells. If the dot-like structure is indeed a focal adhesion-like structure, it is difficult to understand how one large structure in the back of the cell can promote cell migration.

Response: Please note that the here chosen experimental conditions are different from those used to identify focal adhesion-like structures. Fierro Morales et al (*MBoC*, 2025) imaged *Dictyostelium* in development/starvation buffer, whereas amoeba were imaged in our study in full medium, under agar while performing chemotaxis in a folate gradient. We suppose that our conditions, which induce fast migration, do not favor the formation of the cell adhesion structures described before. In fact, this seems to be in line with the conclusion from the Fierro Morales study, which concluded that focal adhesion-like structure act as ‘breaks’ and slow down migration.

To test this hypothesis, we obtained PaxB-deficient cells that were reconstituted with GFP-PaxB. Using these cells, we performed live cell imaging under the same experimental conditions as established for our talin-A experiments. Indeed, we observed the previously described punctate, PaxB-positive adhesion structures (Bukharova et al, *JCS*, 2005; Fierro Morales et al, *MBoC*, 2025) only in very few instances, when cells were hardly migrating. By contrast, cells undergoing cell migration did not show focal adhesion-like structures in our experimental setting and PaxB was rather homogeneously distributed. Interestingly, we observed accumulations of PaxB at the posterior end of migrating cells, similar to what we have described for talin-A. These new images are now included in the manuscript and shown in Extended Data Fig. 6f, g. We also adjusted the text to describe this observation.

3. What is the dynamic range of the talin biosensor? The negative control is performed with LatA, but is there a positive control to know the upper limit?

Response: It is difficult to define an upper limit of tension because the available tools only measure forces up to a certain threshold. Above this threshold, forces cannot be distinguished. However, in response to the reviewer’s remark, we tested whether talin forces may be higher than 6-8 pN, by generating a talin-A tension sensor using the stiffest force sensor peptide monitoring mechanical forces of 9-11 pN. We have

previously shown that mammalian talin-1 is subject to these forces during cell adhesion (Austen et al, *NCB*, 2015). We expressed this new sensor in talin-A-deficient *Dictyostelium* cells and measured FRET efficiencies. Consistent with our hypothesis that talin mechanics are evolutionarily conserved, the data indicate that also talin-A in amoeba is exposed to forces of 9-11 pN. These new data are now shown in Extended Data Fig. 7b.

4. The hypothesis that mammalian cells transition to an integrin-independent amoeboid form of motility is not well-supported. It is clear that when SibA and TalA are expressed in Tln1/2- mammalian cells, the cells move faster, but there is no indication of the type of motility that they are using. Markers and pharmacological inhibitors for amoeboid migration would need to be used to support this statement.

Response: Following the suggestion of the reviewer, we performed additional experiments in which confined cells were treated with para-amino-blebbistatin to inhibit myosin contractility. Previous work by the group of Matthieu Piel demonstrated that this inhibits the fast A2-type of amoeboid migration (Liu et al, *Cell*, 2015). As show in the newly included figures, blebbistatin treatment abrogates SibA/TalA induced movements suggesting that the observed effects are myosin-II-dependent. These data are now shown in the main Fig. 6f, g.

5. The data representation at times is unclear, and the data are not consistently quantified throughout. In several figures, representative images or western blots are shown, and quantification across biological replicates is needed.

Response: We assume that the reviewer refers to the western blots in Fig. 2 and we agree that these important data sets should be quantified. We therefore performed densitometric analyses of the western blots on pY397 FAK and pY118 paxillin and these quantifications are now shown in Fig. 3b, c.

Minor Comments:

-The kymographs are difficult to interpret. What is 0 and 1? I could not find information in the legend or methods.

Response: As shown in Extended Data Fig. 3c, '0' and '1' indicate the start and end of the line that was drawn for the kymograph analysis.

-It would be ideal to better highlight the previous work showing that Dicytostelium do not have FAK and Src homologs. At times, the manuscript is misleading regarding what previous work has already been done in Dictyostelium.

Response: We apologize if previous work was not properly referenced, this was unintentional. We now included more citation to better acknowledge previous work. We currently have 82 references, which is significantly above the suggested number of 50 references.

-What does the TalA tension sensor look like during directed cell migration? Does TalA still form the same large structure in the back of the cell? Does the structure still transmit force during directed cell migration when the cells are migrating at faster speeds?

Response: Please note that the majority of *Dictyostelium* experiments were performed during cell migration, under agar, in a folic acid gradient (Extended Data Fig. 9). We therefore think that the structure is particularly prominent in migrating cells and has a role for anchoring the cell to the underlying substrate in these conditions.

Response to Reviewer #1

The authors responded systematically to all of this reviewer's questions and significantly improved the manuscript.

Regarding point 2 of the discussion negative results of the manuscript, the already confined mechanical status of Dicto cells in stiffness-dependent assays does indeed make it difficult to interpret the data, particularly if excessive confinement triggers cell death (which should be verified by labeling cell death markers) or any other process leading to rounding. This would indeed warrant further research to be published and can be omitted at this stage.

Regarding point 4 of the discussion negative results of the manuscript, this challenging question would certainly warrant further research, which would take a long time, to be conducted as part of future work.

Regarding point 3 of the discussion, the absence of obvious change in the spatial distribution of Tal4 is an interesting result, indicating the absence of detectable membrane Tal4 recycling. Frap experiments at membranes would certainly provide a definitive answer to this question, allowing the potential existence of local recycling helping migration, modifying not a steady state overall distribution at the cellular level. If such an experiment is feasible in the context of the configuration of the present experimental device, it would certainly be nice trying it before publication.

Response: We thank the reviewer again for the constructive suggestions and helpful remarks throughout the review process. We appreciate that the reviewer does not ask for additional experiments at this point and agree that these questions should be addressed in future work.

Response to Reviewer #2

I would like to thank the authors for their thorough and thoughtful responses to my previous comments. I appreciate the significant effort that went into performing the additional experiments, expanding the datasets, and revising the text and figures. The authors have addressed all of my concerns in a clear and comprehensive manner.

I believe the new data and analyses strengthen the manuscript and further support the main conclusions. The presentation is now clear, balanced, and well-structured. I am satisfied with the revisions and have no remaining concerns.

In my opinion, the manuscript is now suitable for publication in Nature Communications.

Response: We would like to thank the reviewer again for the helpful comments and suggestions during the review process. Together with the remarks of the other reviewers, these suggestions have improved the manuscript. Thank you for supporting the publication of the manuscript.

Response to Reviewer #3

This resubmission contains additional data shoring up some of the take-home messages in the original manuscript, however, I'm afraid that some of my main concerns remain.

1. A key point in the manuscript and highlighted in the abstract is that Talin transmits mechanical forces in the absence of an integrin receptor ("we uncover a conserved role for talin in transmitting pN-scale forces into cells, even in unicellular organisms lacking integrin receptors.") This is a provocative statement, but one that I still do not feel is justified by the data. The authors have revised the text to convey the complexity of whether Dicty has a bona fide integrin homologue, and I understand the authors' desire to write such a compelling statement. The talin-centric cell biology is top notch, with beautiful characterization and quantification, but I am not convinced that talin being able to transmit mechanical forces without an integrin is a supported main take-home message of the paper, which will of course limit my overall enthusiasm for the work.

Response: We thank the reviewer for acknowledging the quality of our work and pointing out that the statement on integrin receptors could be perceived as provocative. To be as clear as possible, we adjusted the abstract and the main text to emphasize throughout the manuscript that *Dictyostelium* does not express canonical integrin receptors. Please also note that we have already specified the function of SibA as an integrin equivalent and emphasized the need to clarify the relationship between SibA and integrins.

We do not think that these statements affect the key findings of our study. Please note that we are not arguing that talin transduces forces without engaging transmembrane receptors. In fact, we emphasize that the ability of talin to complex with integrin equivalents like SibA (but potentially also other transmembrane proteins carrying NPXY motifs) is a particularly exciting result of this study. As we have discussed in detail, this mechanism could facilitate cell adhesion in numerous other unicellular eukaryotes that do not express canonical integrins.

We believe that the wording in our manuscript is now unambiguous and fully consistent with the available data on SibA and integrin receptors, and we hope the reviewer will agree.

2. The authors include new data showing PaxB localization to the posterior TalA structures in migrating cells, but there is no quantification to facilitate the interpretation. What percentage of TalA-containing structures also localize PaxB.

Response: Please note that we did not co-express TalA and PaxB in the same cell. We have added the GFP-PaxB experiments to test whether the previously described focal adhesion like-structures form under our experimental conditions. Consistent with the observation in the Fierro Morales et al (*MBoC*, 2025) study, which concludes that these adhesion complexes act as brakes and are associated with stationary or slowly migrating cells, we do not observe focal adhesion like structures in our chemotactically migrating amoeba. As we observed a dot-like accumulation of GFP-PaxB in our system, we found it important to point this out. However, we could not quantify a colocalization with TalA, as

the available TalA antibody only works in western blotting but, at least in our hands, not in immunostainings. To avoid any misunderstandings, we adjusted the discussion to clarify this point.

3. Thank you for including additional experiments to better interpret the talin biosensor dynamic range.

Response: Thank you again for the suggestion; the additional data have improved the manuscript.

4. I'm still having a tough time understanding the Tln1/2- mammalian experiment expressing SibA and/or TalA. This experiment was performed to test whether "the original, mechanical function of talin is sufficient to modulate processes". If the Tln1/2- cells expressing SibA/TalA are indeed undergoing bleb-based myoII-dependent migration (supported by the addition of the new blebbistatin experiments, which are nice), and TalA does not appear to localize in any punctate manner that interacts with ECM, how do these results support a mechanical function or coupling for TalA? Wouldn't the results of this experiment just suggest that TalA can promote bleb-based migration when SibA is present, but not due to adhesion to ECM and force transmission through TalA?

Response: At this point, we can only speculate about how talin force coupling facilitates amoeboid migration in our system. Amoeboid migration occurs in different modes, and most cells can switch between them in order to move forward. Although adhesion forces are believed to play a minor role, the cell surface (plasma membrane) must still engage the intracellular force-generating machinery (actin retrograde flow) in order to squeeze forward.

We currently hypothesize that SibA-TalA facilitates amoeboid migration in our fibroblast model by mediating a mechanical linkage between the plasma membrane and the actin retrograde flow. However, providing direct evidence for this hypothesis would require the generation of additional cell lines, for example to monitor the actin cytoskeleton during live cell experiments or to express mutated TalA, and the use of more sophisticated experimental setups to induce different modes of amoeboid migration. We believe that these time-consuming experiments need to be conducted in future studies, and we hope the reviewer will agree that such experiments are beyond the scope of this study.

5. Thank you for including the requested quantification.

Response: Thank you for the suggestion and for taking the time to carefully evaluate the manuscript.